# Ubiquitin proteolysis of a CDK-related kinase regulates titan cell formation and virulence in the fungal pathogen *Cryptococcus neoformans*

Chengjun Cao [1], Keyi Wang[2], Yina Wang [1], Tong-Bao Liu [1,3], Amariliz Rivera [2] & Chaoyang Xue [1,4,5] ✉

Fungal pathogens often undergo morphological switches, including cell size changes, to adapt to the host environment and cause disease. The pathogenic yeast *Cryptococcus neoformans* forms so-called 'titan cells' during infection. Titan cells are large, polyploid, display alterations in cell wall and capsule, and are more resistant to phagocytosis and various types of stress. Titan cell formation is regulated by the cAMP/PKA signal pathway, which is stimulated by the protein Gpa1. Here, we show that Gpa1 is activated through phosphorylation by a CDK-related kinase (Crk1), which is targeted for degradation by an E3 ubiquitin ligase (Fbp1). Strains overexpressing *CRK1* or an allele lacking a PEST domain exhibit increased production of titan cells similarly to the *fbp1Δ* mutant. Conversely, *CRK1* deletion results in reduced titan cell production, indicating that Crk1 stimulates titan cell formation. Crk1 phosphorylates Gpa1, which then localizes to the plasma membrane and activates the cAMP/PKA signal pathway to induce cell enlargement. Furthermore, titan cell-overproducing strains trigger increased Th1 and Th17 cytokine production in CD4[+] T cells and show attenuated virulence in a mouse model of systemic cryptococcosis. Overall, our study provides insights into the regulation of titan cell formation and fungal virulence.

The maintenance of proper cell size is critical for cellular fitness and function across unicellular and multicellular organisms. Cell size is determined by a number of internal and external factors, including DNA content and nutrients[1–3]. Loss of specific cell size control is often observed in chronic diseases such as cancers[2,3], whereas dramatic cell size change is commonly associated with cellular senescence in human cell lines and budding yeast cells[4]. Pathogenic fungi benefit from their ability to undergo morphological switches and cell size change during infection[5,6]. Elongated hyphal cells are invasive and support tissue penetration, whereas small yeast cells are better suited for dissemination[7,8]. Morphological changes have been studied in major fungal pathogens, including *Candida albicans* and *Aspergillus fumigatus*[6], as well as in endemic fungi, such as *Histoplasma*, *Blastomyces*, and *Coccidioides* spp.[9]. *Candida* species produce three yeast cell types (white, gray and opaque cells) that show different cell sizes and divergences in virulence and mating competence[10]. Although filamentous cells of *Cryptococcus neoformans* are rarely detected during infection[11,12], cellular heterogeneity (cell diameter range from

[1]Public Health Research Institute, New Jersey Medical School, Rutgers University, Newark, NJ 07103, USA. [2]Center for Immunity and Inflammation, New Jersey Medical School, Rutgers University, Newark, NJ 07103, USA. [3] Medical Research Institute, Southwest University, Chongqing 400715, China. [4]Department of Microbiology, Biochemistry and Molecular Genetics, New Jersey Medical School, New Jersey Medical School, Rutgers University, Newark, NJ 07103, USA. [5]Rutgers Center for Lipid Research, Rutgers University, New Brunswick, NJ 08901, USA. ✉e-mail: xuech@njms.rutgers.edu

<1 µm to >100 µm) is often observed during clinical cryptococcal infection[13–15]. Importantly, previous studies indicate that cell size variation during cryptococcal infection shapes fungal pathogenesis in vivo[16,17]. Therefore, the study of *Cryptococcus* cell size heterogeneity offers an excellent system to understand cell size regulation mechanisms.

Previous studies demonstrate that at the early stage of *C. neoformans* pulmonary infection, large encapsulated yeast cells (cell body size over 10 µm) are observed and remain in the lung for extended periods[16,17]. The production of large cells, or cell gigantism in *C. neoformans*, is also known as titan cell formation. Titan cells exhibit increased ploidy with a single nucleus, thicker cell wall, and a large vacuole, and their haploid or aneuploid daughter cells are better adapted to host stressors[17,18]. The regulation of titan cell formation is linked to several signaling pathways[19,20]. The G protein-coupled receptors Ste3α (pheromone receptor) and Gpr5 interact with the Gα protein Gpa1 to activate the cAMP/PKA signaling pathway, which plays an important role in regulating cell size[17,21–25]. Deletions of meiosis-specific genes, such as those encoding the recombinase Dmc1 and the meiotic cohesin complex subunit Rec8, increase the proportion of titan cells, indicating the involvement of meiosis machinery in ploidy reduction during titan cell proliferation[20]. The transcription factors Usv101, Pdr802 and cyclin protein Cln1 also regulate titan cell production[24,26,27]. Despite these valuable observations, our understanding of regulatory mechanisms that control titan cell production remains incomplete. It is highly likely that novel regulatory elements that support fine-tuned cell size regulation in *C. neoformans* have yet to be discovered.

The ubiquitin-proteasome system (UPS) is a principal mechanism for controlled protein degradation in eukaryotic cells and regulates critical cellular functions. UPS has emerged as an attractive drug target for human diseases[28,29] and requires the concerted action of an E1 ubiquitin-activating enzyme, an E2 ubiquitin-conjugating enzyme, and an E3 ubiquitin ligase[30]. The SCF (Skp1, Cullins, F-box proteins) E3 ligase represents the largest E3 ubiquitin ligase family whose exchangeable F-box subunits enable SCF to target specific protein substrates for ubiquitination[30,31]. Despite their universal importance, the regulation of cellular function in fungal pathogens, including *C. neoformans*, by UPS remains understudied. Our previous studies determined that Fbp1, a F-box protein, is critical for *Cryptococcus* virulence[32,33]. The *fbp1Δ* mutant persists at low levels in the lung without causing systemic dissemination in a murine model of cryptococcosis[33,34]. Additionally, Fbp1 is essential for fungal meiosis and sporulation as the *fbp1Δ* mutant fails to produce basidiospores[32].

In this study we set out to uncover the mechanism of low pulmonary persistence seen in the *fbp1Δ* mutant. We determined that Fbp1 negatively regulates *C. neoformans* cell size through its substrate Crk1, a CDK-related kinase with a known function in meiosis regulation. Overexpression of the *CRK1* wild type, or an allele lacking a PEST domain, increased the proportion of titan cells, similar to *fbp1Δ*. In contrast, *CRK1* deletion decreased the population of titan cells. We found that loss of Fbp1 leads to Crk1 accumulation, which in turn regulates the phosphorylation and plasma membrane localization of Gpa1. The Gpa1-cAMP/PKA pathway activation promoted fungal cell size enlargement. We also observed that a *Cryptococcus* strain with overly-accumulated Crk1 attenuated fungal virulence and enhanced protective immune response in the host. Overall, our study reveals a regulatory mechanism of cell size change that is controlled by the SCF(Fbp1) E3 ligase-mediated UPS and the Gpa1 G protein signaling pathway through a kinase regulator, Crk1.

## Results

### Persistent *fbp1Δ* cells in infected lungs contain a higher percentage of titan cells

Fbp1 is required for fungal virulence in *C. neoformans*[32]. Although the *fbp1Δ* mutant does not cause disseminated cryptococcosis, a small population of live yeast cells persist in the *fbp1Δ*-infected lung[33,34]. To characterize the residual *fbp1Δ* cell population in the lung, lungs infected with wild type strain H99, the *fbp1Δ* mutant, or its complemented strain (*fbp1Δ+FBP1*) were examined using Haematoxylin and Eosin (H&E) staining at 3-, 7-, or 15-days post-inoculation and at the end time point (ETP) of the experiment (Supplementary Fig. 1a). We found that ~45% of fungal cells in *fbp1Δ*-infected lungs were larger than 10 µm in diameter throughout the course of infection. In contrast, the percentage of large cells decreased from 25% at day 3 to 8% at ETP in H99- or *fbp1Δ+FBP1*-infected lungs (Fig. 1a). Consistent with these findings, we analyzed the bronchoalveolar lavage fluids (BALF) of mice infected with $5 \times 10^6$ CFU of H99, *fbp1Δ* or *fbp1Δ+FBP1* strain at 3 days post-infection and found that approximately 21% of recovered H99 or *fbp1Δ+FBP1* cells and more than 55% of recovered *fbp1Δ* cells measured over 10 µm in cell body size (Fig. 1b–d).

To better characterize the nature of persistent *fbp1Δ* mutant cells in the lung and further analyze the potential function of Fbp1 in cell size regulation, we analyzed titan cell production of the *fbp1Δ* mutant and H99 under in vitro titan cell inducing conditions following the protocol optimized by Hommel et al.[24]. The large cells induced in vitro contained a large vacuole, single nucleus, and increased DNA content, hallmark features of titan cells (Fig. 1e and Supplementary Fig. 1b–d). Although both H99 and the *fbp1Δ* mutant can be induced to produce titan cells, we observed a higher percentage of titan cells (48.7% in *fbp1Δ* vs. 22.8% in H99 and 19.6% in *fbp1Δ+FBP1*) and a larger median cell body size in the *fbp1Δ* mutant compared to H99 (Fig. 1f–h). These results indicate that Fbp1 regulates titan cell formation in *C. neoformans*.

### Crk1 is a substrate of Fbp1 and is required for sporulation

Meiosis-specific regulators Dmc1 and Rec8 are reportedly required for the ploidy reduction of titan cell proliferation during infection[20]. We have previously shown that *fbp1Δ* cells, similar to *dmc1Δ* and *rec8Δ* mutants, fail to undergo sporulation or produce basidiospores, despite producing normal dikaryotic hyphae and basidia[32]. These observations suggest that Fbp1 may contribute to depolyploidization of titan cells, likely through the regulation of meiosis machinery. Therefore, we posited that some Fbp1 substrates required for meiosis regulation may be involved in titan cell production. We further hypothesized that Crk1 (CNAG_06193), a CDK-related kinase and a meiosis regulator that interacts with Fbp1[32], is a potential Fbp1 substrate involved in Fbp1-mediated meiosis regulation and cell size control. Structural analysis showed that Crk1 contains a protein kinase domain and a putative PEST region (based on the ePESTFind program, http://emboss.bioinformatics.nl/cgi-bin/emboss/epestfind) which is rich in proline (P), glutamic acid (E), serine (S), and threonine (T) (Fig. 2a). The PEST domain localizes to the C-terminus of Crk1 and is a signature for protein degradation by ubiquitin-proteasome system[33,35].

To test these hypotheses, we first reexamined the interaction between Fbp1 and Crk1 using a yeast two-hybrid assay and found that the C-terminus of Crk1 was sufficient to support the interaction with Fbp1 (Fig. 2b). The interaction between Crk1 and Fbp1 was further validated through a co-immunoprecipitation (co-IP) assay in which both Fbp1:FLAG and Crk1:HA could be detected in the product immunoprecipitated using FLAG antibody from the total protein of a strain expressing both Fbp1:FLAG and Crk1:HA, but not from the total protein of strains expressing either Fbp1:FLAG or Crk1:HA (Fig. 2c). We also detected an interaction between Crk1 and Fbp1 without the F-box domain (Fbp1^ΔF) (Fig. 2b, c), suggesting that this interaction in *C. neoformans* is independent of the Fbp1 F-box domain.

If Crk1 is a substrate of Fbp1, Crk1 would be more stable in the *fbp1Δ* background due to its lack of ubiquitination and subsequent degradation. Therefore, we expressed Crk1:HA under the control of the native *CRK1* promoter and examined Crk1 abundance in both H99 and *fbp1Δ* backgrounds. We observed that Crk1:HA was more

abundant in the *fbp1*Δ mutant than in H99 (Fig. 2d). To examine whether the stability of the Crk1 protein depended on Fbp1, we expressed the Crk1:HA protein under the control of an inducible *CTR4* promoter in the H99 and *fbp1*Δ backgrounds. *CRK1:HA* gene expression was then induced by adding the copper chelator BCS and stopped by adding copper in order to monitor the stability of existing proteins in the cell. We observed that the Crk1:HA protein was degraded in a time-dependent manner in the H99, however it was stable in the *fbp1*Δ mutant (Fig. 2e). Fbp1 failed to degrade the Crk1 allele that lacked the PEST region ($P_{CTR4}$-*CRK1*$^{\Delta PEST}$:*HA*) and Crk1$^{\Delta PEST}$:HA was stable in H99 over the observed time period, similar to the stability noted for Crk1$^{\Delta PEST}$ in the *fbp1*Δ background strain (Fig. 2f).

SCF ligase recruits substrates for ubiquitination[30,31], therefore we tested whether Crk1 ubiquitination is dependent on Fbp1. *CRK1:HA* was overexpressed in the H99 and *fbp1*Δ backgrounds, respectively. The cells were cultured for 4 h in YPD in the presence or absence of MG132, a cell-permeable proteasome inhibitor that stabilizes poly-ubiquitinated proteins[36]. Crk1 was purified from total proteins via HA antibody immunoprecipitation and putatively ubiquitinated Crk1 was

detected as a ladder-like smear with high-molecular-weight protein in a western blot using HA antibody. We observed that the ubiquitination level of Crk1 is diminished in the *fbp1*Δ mutant compared to the H99 background while in the presence of MG132 (Fig. 2g). Additionally, we detected small amounts of polyubiquitinated Crk1 in both H99 and the *fbp1*Δ mutant even in the absence of MG132, suggesting ubiquitinated Crk1 may mediate non-proteolytic functions[37]. Together, these data indicate that Crk1 interacts with Fbp1 through its C-terminal region and that the stability and ubiquitination of the Crk1 protein is dependent on Fbp1, confirming that Crk1 is a substrate of Fbp1 in *C. neoformans*.

In a bilateral mating setting, the *crk1*Δ mutant and the *CRK1* overexpression strains (*CRK1*$^{OE}$), in which a GFP:Crk1:HA fusion protein expression is under the control of Histone H3 promoter ($P_{HIS}$-*GFP:CRK1:HA*), showed a significant sporulation defect with reduced basidiospore production, suggesting that proper expression of *CRK1* is necessary for sporulation in *C. neoformans* (Fig. 2h and Supplementary Fig. 2). DAPI staining of nuclei in mating hyphae showed a defect in nuclear division during meiosis in both the *crk1*Δ mutant and *CRK1*$^{OE}$ strain, indicating that Crk1 is required for proper regulation of fungal

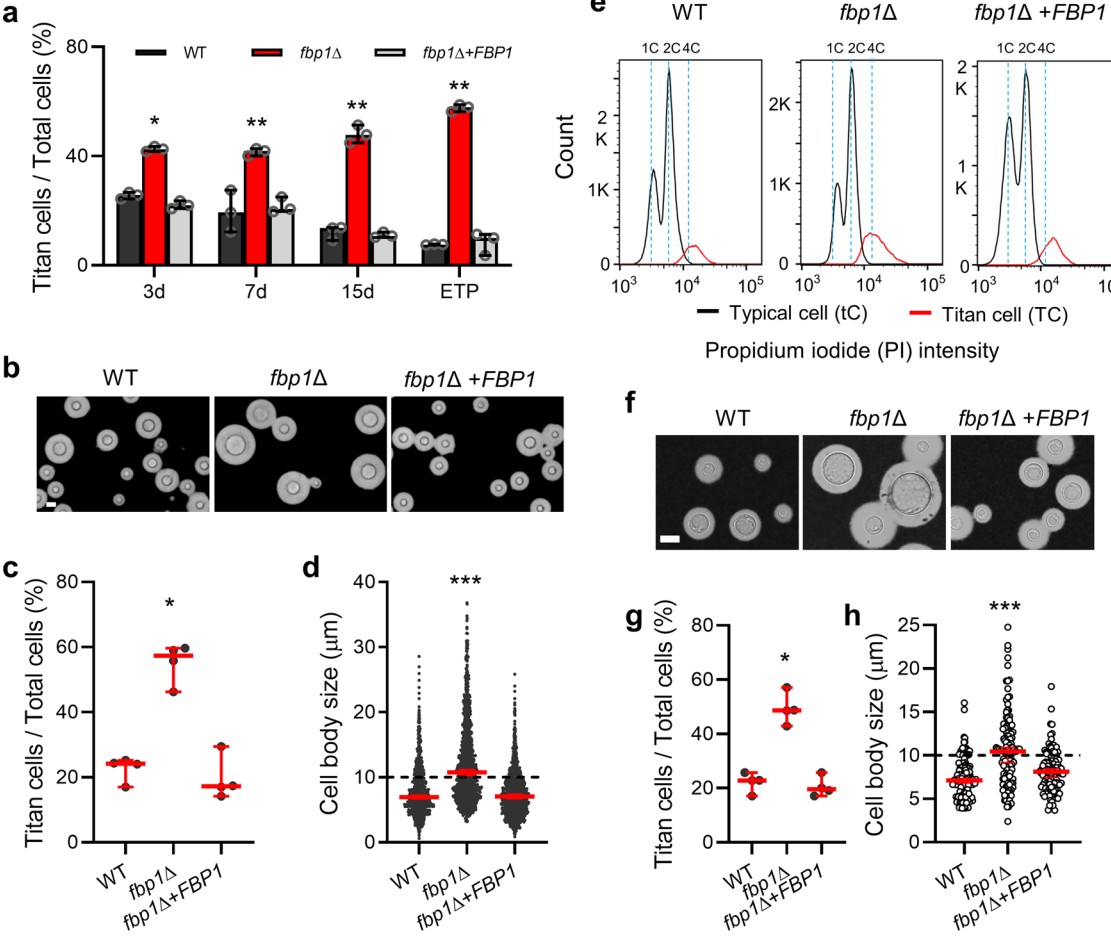

**Fig. 1 | The E3 ligase Fbp1 is required for cell size control in *C. neoformans*.**
**a** Percentage of titan cell in the lungs during *Cryptococcus* infection. Error bar indicates 95% confidence interval of the median for 3 independent experiments. Statistical analysis for all measurements in this figure was performed with the two-sided Kruskal-Wallis nonparametric test for multiple comparisons. *$P = 0.031$; **$P = 0.003$; 0.009; 0.001. "ETP", end time point. **b** Representative images of *C. neoformans* in BALF collected from the wild type H99-, *fbp1*Δ-, or *fbp1*Δ+*FBP1*-infected lungs after 3 days post-infection. Bar, 10 μm. **c, d** Quantitative measurement of titan cell percentage (**c**) and cell body size (**d**) in BALF. The data shown are cumulative from four mice per group and the pooled dataset across four mice ($n = 1500$ cells) are shown for cell size measurements. Error bar indicates 95% confidence interval of the median. *$P = 0.031$;

***$P < 0.0001$. **e** FACS analysis of DNA content in H99, *fbp1*Δ, and *fbp1*Δ+*FBP1* strains. Cells were fixed and stained by propidium iodide (PI) after 3 days of incubation in titan cell inducing conditions. The population of large cells showed increased PI fluorescence intensity to >2 C, whereas the cells of typical size population harbored 1 C or 2 C PI intensity. **f** Representative images of H99, *fbp1*Δ, and *fbp1*Δ+*FBP1* strains under in vitro titan cell inducing conditions. Bar, 10 μm. **g, h** Quantitative measurement of titan cell percentage (**g**) and cell size (**h**) of these strains under in vitro titan cell inducing conditions. The titan cell percentage data are cumulative from four independent experiments and the cell size data ($n = 100$ cells) are representative of four independent experiments. Error bar indicates 95% confidence interval of the median. *$P = 0.030$; ***$P < 0.0001$. Source data are provided as a Source Data file.

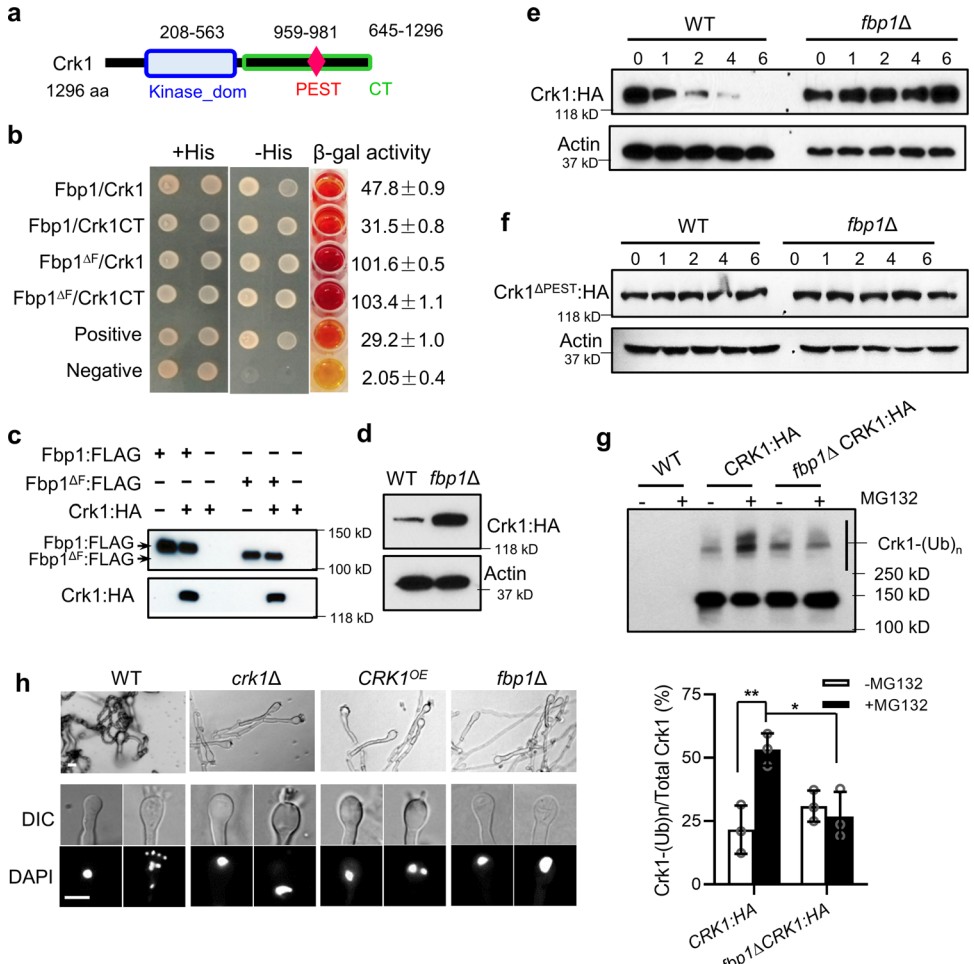

**Fig. 2 | Crk1 is a substrate of Fbp1 required for meiosis. a** A schematic illustration of Crk1 proteins in *C. neoformans*. aa amino acids, Kinase_dom protein kinase domain, PEST PEST domain, CT C-terminus. **b** The interaction between Fbp1 and Crk1 in a yeast two-hybrid interaction assay. Colonies were tested on media lacking histidine. β-galactosidase activity assays were performed to verify the interaction. The assay was repeated at least three times with similar outcomes. Fbp1$^{\Delta F}$, Fbp1 lacking the F-box domain. Crk1CT, C-terminus of Crk1. **c** Lysates were prepared from strains expressing either Fbp1:FLAG or Fbp1$^{\Delta F}$:FLAG, Crk1:HA, or both, and immunoprecipitated using anti-FLAG affinity gel, then analyzed by immunoblotting using anti-FLAG (top) or anti-HA (bottom). The data are representative of three independent experiments. **d** Cells of indicated genetic background expressing $P_{CRK1}$-*CRK1:HA* were harvested after being cultured in YPD overnight and the abundance of Crk1 was monitored by immunoblotting. The data are representative of three independent experiments. **e** The $P_{CTR4}$-*CRK1:HA* construct was expressed in both the WT and *fbp1Δ* strains. The level of Crk1:HA was determined by immunoblotting using anti-HA at indicated time points (in hours) after blocking *CRK1* transcription by copper. The abundance of Actin protein detected by anti-Actin antibody was used as a loading control. The data are representative of three

independent experiments. **f** The stability of Crk1$^{\Delta PEST}$ protein in the WT and *fbp1Δ* backgrounds. The $P_{CTR4}$-*CRK1$^{\Delta PEST}$:HA* construct was expressed in the WT and *fbp1Δ* backgrounds and the level of Crk1$^{\Delta PEST}$:HA was determined by immunoblotting using anti-HA at indicated time points after blocking *CRK1* transcription by copper. The protein detected by anti-Actin antibody was used as a loading control. The data are representative of three independent experiments. **g** Accumulation of Crk1 polyubiquitination (Crk1-(Ub)$_n$) was detected using HA antibody. Crk1:HA were expressed in the WT and *fbp1Δ* backgrounds. Overnight cultures were collected and total proteins were extracted. Crk1:HA was purified by HA antibody immunoprecipitation and detected using anti-HA. Error bar indicates 95% confidence interval of the median for three independent experiments. Statistical analysis was performed based on two-sided Mann–Whitney test. *$P = 0.017$; **$P = 0.009$. **h** Crk1 is required for meiosis and sporulation. Bilateral mating assays for wild type, the *crk1Δ* mutant, the *CRK1* overexpression strain (*CRK1$^{OE}$*), and the *fbp1Δ* mutant. Mating structures were photographed after 7 days of incubation in the dark at 25 °C on MS medium. Images are representative of three independent experiments. Bars, 5 μm. Source data are provided as a Source Data file.

meiosis[38,39]. Although reduced, the *CRK1$^{OE}$* strains do produce spores, compared to a complete lack of spore formation observed in *fbp1Δ* bilateral mating. These observations suggested that Crk1 plays an important role in Fbp1-mediated fungal meiosis and sporulation and may be involved in titan cell production in *C. neoformans*.

**Crk1 regulates Fbp1-mediated cell size change**

To evaluate the potential involvement of Crk1 in Fbp1-mediated titan cell regulation, we examined titan cell production in the *crk1Δ* mutant, the *CRK1$^{OE}$* ($P_{HIS}$-*GFP:CRK1:HA* or $P_{ACT1}$-*CRK1:mCherry*) strain, and the *CRK1* complement strain (*crk1Δ+CRK1*) under in vitro titan cell inducing conditions as previously reported[24]. We observed a higher

percentage of titan cells and a larger median cell body size in the *CRK1$^{OE}$* strain (44.8%, median cell size 9.6 μm) and reduced titan cell proportion and median cell size in the *crk1Δ* mutant (2.8%, median cell size 6.8 μm) compared to H99 (22.5%, median cell size 8.1 μm) or *crk1Δ* +*CRK1* (23.0%, median cell size 7.8 μm) strains (Fig. 3a, b). Large cells of the *CRK1$^{OE}$* strain were polyploid with a single nucleus (Fig. 3c and Supplementary Fig. 3). Due to the potential importance of the PEST domain for Crk1 protein stability, we generated strains overexpressing *CRK1* lacking the PEST domain (*CRK1$^{\Delta PEST}$*), in which a *CRK1$^{\Delta PEST}$:mCherry* fusion protein expression is under the control of Actin promoter ($P_{ACT1}$-*CRK1$^{\Delta PEST}$:mCherry*). We examined the DNA content of large cells and measured titan cell percentage and median cell body size of the

$CRK1^{\Delta PEST}$ overexpressing strain (Fig. 3a–c). The $CRK1^{\Delta PEST}$ strain (54.6%, median cell size 11.1 μm) produced more titan cells and larger size cells at levels comparable to the $CRK1^{OE}$ strain.

We isolated BALF from mice infected with different strains and analyzed their titan cell production to examine the role of Crk1 in cell size control during infection (Figs. 3d–f). Compared to 24.2% of titan cells in the BALF from H99 infected mice, we detected significantly fewer titan cells in the crk1Δ infected mice (10.5%) and higher proportions of titan cells in $CRK1^{OE}$ (48.3%) and $CRK1^{\Delta PEST}$ (66.2%) infected mice (Fig. 3e), which is consistent with the observations of in vitro conditions. Accordingly, the median cell body size was larger in $CRK1^{OE}$ (9.3 μm) and in $CRK1^{\Delta PEST}$ (13.2 μm) strains and was smaller in the crk1Δ mutant (6.1 μm) during infection compared to the H99 (6.9 μm) or crk1Δ+CRK1 (7.4 μm) strains (Fig. 3f). Additionally, we observed that the $CRK1^{\Delta PEST}$ strain produces more titan cells than the $CRK1^{OE}$ strain under both in vivo and in vitro titan cell inducing conditions, which is consistent with protein levels of Crk1 in both strains (Supplementary Fig. 4). Given that both the $CRK1^{\Delta PEST}$ and $CRK1^{OE}$ strains had similar RNA levels of CRK1 (Supplementary Fig. 4a), Crk1 was more stable in $CRK1^{\Delta PEST}$ than in the $CRK1^{OE}$ background (Supplementary Fig. 4b, c). Altogether, these data indicate that Crk1 is involved in Fbp1-mediated cell size regulation.

To understand the global gene expression landscape of Fbp1 and Crk1 during titan cell production, we performed RNA-Seq to compare transcriptomic profiles of the WT, fbp1Δ, $CRK1^{OE}$, and $CRK1^{\Delta PEST}$ strains under titan cell inducing conditions (Fig. 3g–i and Supplementary Data 1). Correlation coefficient analyses demonstrated that fbp1Δ, $CRK1^{OE}$, and $CRK1^{\Delta PEST}$ strains exhibited largely overlapping global gene expression patterns with some distinctions (Fig. 3g). Compared to H99, there are more differentially-expressed genes (DEGs), and particularly unique DEGs, in the $CRK1^{OE}$ strain (1007 genes) than in fbp1Δ and $CRK1^{\Delta PEST}$ strains (715 and 774 genes, respectively) (Fig. 3h), indicating that Crk1 likely retains roles other than those that function as a substrate of Fbp1 under titan cell inducing conditions. The fbp1Δ, $CRK1^{OE}$, and $CRK1^{\Delta PEST}$ strains shared 247 common DEGs compared to H99 (Fig. 3h). Additionally, these overlapping genes possess an identical regulation pattern where 190 genes are upregulated and 57 genes are downregulated across all three comparison groups (fbp1Δ vs WT, $CRK1^{OE}$ vs WT and $CRK1^{\Delta PEST}$ vs WT) (Supplementary Data 1). Gene clustering and category (GO term) analysis of these DEGs showed over-representation of genes related to metabolic processes, transmembrane transporters, protein processing, as well as signaling regulation (Fig. 3i). A large number of genes that encode chaperone and heat-shock proteins were upregulated, including the gene encoding calnexin (CNAG_02500), which is reportedly involved in cell size regulation[25]. We also evaluated the predicted interaction network of 247 common genes using the STRING[40] (Search Tool for Retrieval of Interacting Genes/Proteins) database (http://string-db.org) (Supplementary Fig. 5). The greatest number of interactions occurred among proteins associated with chaperone and heat-shock proteins, consistent with the Gene Ontology analysis (Supplementary Data 1). A second interaction cluster included ribosome, spliceosome, DNA repair and DNA replication associated with pre-mRNA splicing and cell cycle control. Additional clusters included interactions among proteins involved in lipid metabolism, sugar metabolism, and other functions (Supplementary Fig. 5). Although the meiosis-related genes that are previously reported to be involved in titan cell production (such as DMC1, REC8, and SPO11) were not highly regulated in our RNA samples, we found several genes involved in meiosis and the cell cycle that were differentially-regulated, suggesting their potential role in cell size regulation (Supplementary Data 1). These included SNF1 (CNAG_06552, encoding a CAMK/CAMKL/AMPK protein kinase), SKS1 (CNAG_06568, encoding a serine/threonine protein kinase), TOR1

(CNAG_06642, encoding an atypical/PIKK/FRAP protein kinase), and genes encoding DNA repair proteins (CNAG_04733 and CNAG_02512). Detailed analyses of DEGs are presented in Supplementary Data 1.

## Crk1 regulates titan cell formation via Gpa1 G protein signaling

Previous studies show that Gpa1 G protein signaling is essential for mating and titan cell production in *C. neoformans*[19,21,41,42]. The strain expressing the GPA1 dominant active allele ($GPA1^{Q284L}$) showed reduced sporulation and increased titan cell production[21,43], similar to the $CRK1^{OE}$ strain and the fbp1Δ mutant, suggesting that the Gpa1 signaling pathway may be involved in Crk1-mediated regulation of titan cell formation in *C. neoformans*. To test this hypothesis, we examined the expression of protein kinase A gene PKA1, a key component of the Gpa1-cAMP pathway, in wild type H99, the crk1Δ mutant and the $CRK1^{OE}$ strain under in vitro titan cell inducing conditions. The expression of PKA1 was significantly downregulated in the crk1Δ mutant compared to other strains (Supplementary Fig. 6a).

cAMP is a key second messenger in Gpa1 signaling. Previous studies demonstrate that Gpa1 activation increases production of intracellular cAMP levels, which activates downstream protein kinase A Pka1 in *C. neoformans*[44,45]. To better understand the potential role of kinase Crk1 in Gpa1 signaling regulation, we measured titan cell production in H99 and the crk1Δ mutant in the presence or absence of exogenous cAMP in the medium. We observed that the addition of exogenous cAMP significantly increased titan cell production with increased DNA content in the crk1Δ mutant (Fig. 4a and Supplementary Fig. 6b, c). Meanwhile, we found that the intracellular cAMP level was significantly lower in the crk1Δ mutant and higher in the $CRK1^{OE}$ and $CRK1^{\Delta PEST}$ strains compared to the H99 in a cAMP assay (Fig. 4b). These results demonstrate that Crk1 is required for the activation of Gpa1-cAMP signaling. In a genetic epistasis analysis, we found that over-expression of CRK1 in the gpa1Δ mutant background (gpa1Δ $CRK1^{OE}$) failed to restore defective titan cell formation in the gpa1Δ mutant (Fig. 4c and Supplementary Fig. 6d). Expression of the $GPA1^{Q284L}$ allele in a crk1Δ mutant background (crk1Δ $GPA1^{Q284L}$) yielded significantly enlarged cells, similar to the wild type expressing $GPA1^{Q284L}$ (Fig. 4c and Supplementary Fig. 6d). All these enlarged cells exhibited increased DNA content compared to typical cells (Supplementary Fig. 6d). These data indicate that Crk1 likely functions upstream of Gpa1 and positively regulates Gpa1-cAMP signaling-mediated titan cell production.

## Crk1 interacts with Gpa1 and regulates Gpa1 phosphorylation and localization

Phosphorylation of the Gα subunit of heterotrimeric G protein complexes is required for the proper activation of G protein signaling[46]. Since Crk1 is a serine/threonine kinase, we hypothesize that Crk1 regulates Gpa1 through Gpa1 phosphorylation. We first tested potential interaction between Gpa1 and Crk1 in a yeast two-hybrid system and found that both Gpa1 and Gpa1^{Q284L} interacts with Crk1 (Fig. 4d). The interaction was confirmed by a Co-IP assay using a H99 strain expressing both Gpa1:FLAG and Crk1:HA. Gpa1:FLAG was detected in the product immunoprecipitated using HA antibody in a western blot (Fig. 4e). To determine whether Crk1 accounts for Gpa1 phosphorylation in *C. neoformans* during titan cell formation, H99, crk1Δ, crk1Δ+CRK1, $CRK1^{OE}$, $CRK1^{\Delta PEST}$, and fbp1Δ strains expressing Gpa1:FLAG were generated and cultured in titan cell inducing conditions or nitrogen starvation conditions. Nitrogen starvation conditions are reported to trigger phosphorylation of *S. cerevisiae* Gpa2[47], a homolog of Gpa1 in *C. neoformans*[42].

It has been reported that protein phosphorylation alters the electrophoretic mobility of the protein when resolved by SDS-polyacrylamide gel electrophoresis (SDS-PAGE)[48,49]. We extracted total proteins from H99 expressing Gpa1:FLAG under titan cell inducing conditions and examined Gpa1 phosphorylation in a western blot

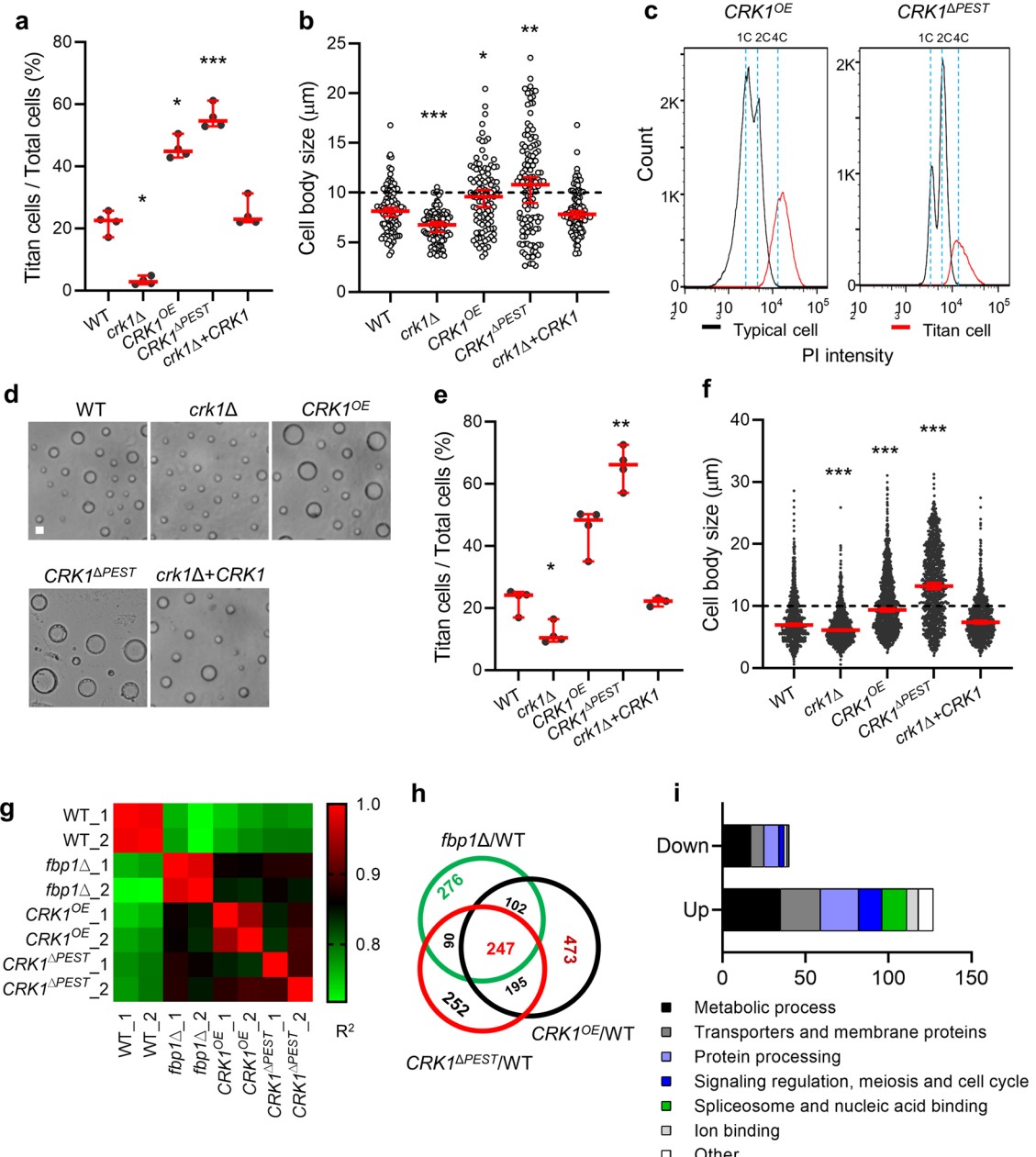

**Fig. 3 | Deletion of Fbp1 increases the proportion of titan cells through Crk1 protein accumulation. a, b** Quantitative measurement of titan cell proportion (**a**) and cell size (**b**) under in vitro titan cell inducing conditions. The titan cell percentage data are cumulative from four independent experiments and the cell size data ($n = 100$ cells) are representative of four independent experiments. Error bar indicates 95% confidence interval of the median. Statistical analysis was performed with the two-sided Kruskal-Wallis nonparametric test for multiple comparisons. *$P = 0.05$; 0.03; ***$P = 0.0004$ in panel (**a**). *$P = 0.01$; **$P = 0.001$; ***$P < 0.0001$ in panel (**b**). **c** FACS analysis of DNA content in the $CRK1^{OE}$ and the $CRK1^{\Delta PEST}$ strains. Cells were fixed and stained by propidium iodide (PI) after 3 days of incubation in titan cell inducing conditions. The population of large cells showed increased PI fluorescence intensity to >2 C, while the cells of typical size population harbored 1 C or 2 C PI intensity. **d** Representative images of cells in BALF collected from a mouse infected with WT, $crk1\Delta$, $CRK1^{OE}$, $CRK1^{\Delta PEST}$, or a $crk1\Delta+CRK1$ strain after 3 days post-infection. Bar, 10 μm. **e, f** Titan cell percentage (**e**) and cell size (**f**) in BALF. The data shown for titan cell percentage are cumulative from four mice and the data shown for cell size are cumulative from 1500 cells. Error bar indicates 95% confidence interval of the median. Statistical analysis was performed with the two-sided Kruskal-Wallis nonparametric test for multiple comparisons. *$P = 0.024$; **$P = 0.002$ in panel (**e**). ***$P < 0.0001$ in panel (**f**). **g** Correlation coefficient of the gene expression profiles in WT, $fbp1\Delta$, $CRK1^{OE}$, and $CRK1^{\Delta PEST}$ strains that were cultured under titan cell inducing conditions at 30 °C for 3 days. The cells were then collected and total RNA was extracted. Clustering analysis of different samples was performed using the R package plots v3.1.3 (https://cran.r-project.org/web/packages/gplots/). The consistency between different samples in the matrix is indicated by different colors. Colors ranging from green to red (levels 0.7–1) indicate an increase in consistency. $R^2$: Square of Pearson correlation coefficient (R). **h** Venn diagram showing numbers of differentially expressed genes in three comparison groups ($fbp1\Delta$ vs WT, $CRK1^{OE}$ vs WT, and $CRK1^{\Delta PEST}$ vs WT) and the overlap between these differentially-expressed genes among three mutant strains. A 2-fold cutoff ($\log_2(x) = 1$) between two different strains was used to define differentially-expressed genes. **i** Categories of 247 differentially expressed genes overlapped in these three groups based on GO term. Source data are provided as a Source Data file.

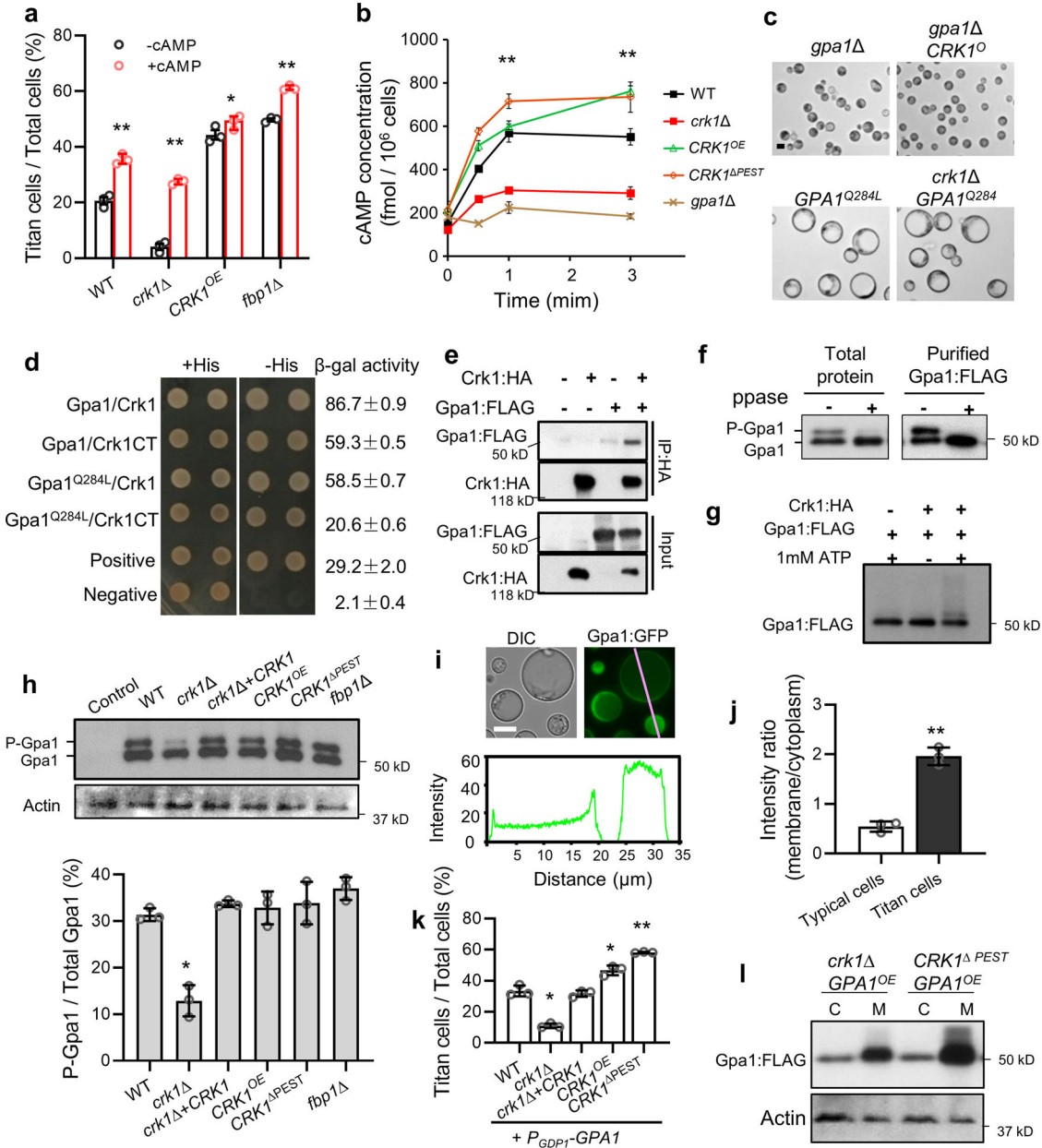

**Fig. 4 | Crk1 regulates titan cell formation through the Gpa1-cAMP signaling pathway. a** Titan cell percentage under in vitro titan cell inducing conditions in the absence or presence of 10 mM extracellular cAMP. Error bar indicates 95% confidence interval of the median for 3 independent experiments. Statistical analysis was performed based on two-sided Mann−whitney test. *$P = 0.030$, **$P = 0.02$. **b** Measurement of cAMP production in response to glucose. Statistical analysis was performed with the Kruskal−Wallis nonparametric test for multiple comparisons. **$P < 0.0001$. **c** Representative cell images of the indicated strains after incubated in minimal medium (MM) for 3 days at 30 °C. Bar, 5 μm. **d** The interaction between Crk1 and Gpa1 in a yeast two-hybrid interaction assay. Gpa1$^{Q284L}$, Gpa1 dominant active allele; Crk1CT, C-terminus of Crk1. **e** Co-immunoprecipitation assay. Lysates from strains expressing either Gpa1:FLAG or both Gpa1:FLAG and Crk1:HA, and immunoprecipitated using anti-HA, then analyzed by immunoblotting using anti-FLAG and anti-HA (IP). **f** Total protein extracts or purified Gpa1:FLAG from the strain expressing $P_{GDP1}$-GPA1:FLAG (CUX1196) were treated with protein phosphatase (ppase) and Gpa1 signal was detected using anti-FLAG. **g** In vitro kinase assay was perform for purified Gpa1:FLAG that was pre-treated with protein phosphatase (ppase), and then mixed with purified Crk1-HA in the absence (−) or presence (+) of ATP. The Gpa1 was detected using anti-FLAG. **h** Gpa1 phosphorylation in titan cell inducing conditions. Total protein from strains expressing Gpa1:FLAG was analyzed by western blotting with an FLAG antibody (top) and the abundance of

phosphorylated Gpa1 (P-Gpa1) protein in the indicated strains was quantified (bottom). Error bar indicates 95% confidence interval of the median for three independent measurements. Statistical analysis was performed with the two-sided Kruskal-Wallis nonparametric test for multiple comparisons. *$P = 0.043$. **i** Representative images to show Gpa1 localization under titan cell inducing conditions (top), and fluorescence intensity plot along a cellular axis indicated with a white line on the image (bottom). Bars, 10 μm. **j** Quantitative measurement of fluorescence signals using ImageJ. Error bar indicates the standard deviation of the mean. Statistical analysis was performed based on two-sided Mann−whitney test. **$P = 0.002$. **k** Quantitative measurement of titan cell proportion in cells over-expressing *GPA1* under in vitro titan cell inducing conditions. Error bar indicates 95% confidence interval of the median for 3 independent experiments. Statistical analysis was performed with the two-sided Kruskal-Wallis nonparametric test for multiple comparisons. *$P = 0.024$; $0.035$; **$P = 0.0004$. **l** Under titan cell inducing conditions, Cytosolic (C) and membrane (M) fractions were separated from cells of the *crk1Δ* and *CRK1$^{ΔPEST}$* strains expressing $P_{GDP1}$-GPA1:FLAG represent typical cells (*crk1Δ GPA1$^{OE}$*) and titan cells (*CRK1$^{ΔPEST}$ GPA1$^{OE}$*), respectively. Gpa1:FLAG was detected using anti-FLAG. The data or images shown in this figure are all representative of three independent experiments with similar results. Source data are provided as a Source Data file.

using anti-FLAG antibody. Indeed, we observed differential mobility of Gpa1 by immunoblotting. To confirm that Gpa1 phosphorylation is responsible for the mobility shift, we treated the total protein extracts or the purified Gpa1:FLAG with protein phosphatase. We detected the loss of a slower migrating signal of Gpa1 in the phosphatase treated samples (Fig. 4f), suggesting that variations in motility during electrophoretic analysis are likely due to changes in phosphorylation. To test whether Crk1 is capable of directly phosphorylating Gpa1, we purified Crk1:HA and Gpa1:FLAG and performed an in vitro kinase assay. Total protein was collected from overnight YPD cultures of H99 expressing Gpa1:FLAG and treated with protein phosphatase to remove phosphorylation before purification. We observed the slower migrating form of Gpa1 only in fractions treated subsequently with both ATP and Crk1:HA (Fig. 4g), indicating that Crk1 directly phosphorylates Gpa1.

Next, we compared Gpa1 phosphorylation in different strain backgrounds. Total protein from the aforementioned strains (H99, *crk1Δ*, *crk1Δ+CRK1*, *CRK1^OE^*, *CRK1^ΔPEST^*, and *fbp1Δ* strains expressing Gpa1:FLAG) under different culture conditions was isolated and analyzed for Gpa1 phosphorylation. We observed diminished phosphorylation of Gpa1 in cells lacking Crk1 compared to H99 under titan cell inducing conditions (Fig. 4h). There were no significant differences in Gpa1 phosphorylation level between *crk1Δ* and other strains when cultured in either nitrogen starvation conditions (Supplementary Fig. 6e) or YPD rich medium (Supplementary Fig. 6f), indicating that Crk1 is required for Gpa1 phosphorylation during the induction of titan cell formation, but not under other testing conditions.

Previous studies show that Gpa1 interacts with the receptors Ste3 and Gpr5 to activate the Gpa1-cAMP signal pathway, indicating plasma membrane localization of Gpa1 during titan cell production[16,17]. Studies in *S. cerevisiae* report that changes in G protein phosphorylation levels altered their subcellular localization[47]. Hence, we investigated the subcellular localization of GFP-fused Gpa1 expressed from the *ACT1* promoter in H99, *crk1Δ*, and *crk1Δ+CRK1* strain backgrounds. Although the Gpa1:GFP signal was detected in both the cytosol and plasma membrane in all strain backgrounds (Supplementary Fig. 6g), our data showed that the GFP signal in titan cells was enriched on the plasma membrane, whereas the signal was primarily concentrated in the cytosol in typical cells (Fig. 4i, j). Consistent with this observation, the *crk1Δ* mutant produced significantly fewer titan cells than H99 or the *crk1Δ+CRK1* strain when *GPA1* was overexpressed (Fig. 4k and Supplementary Fig. 6h). In contrast, the *CRK1^OE^* and *CRK1^ΔPEST^* strains produced more titan cells than H99. Since the *CRK1^ΔPEST^* strain produced more titan cells than did the *crk1Δ* mutant when overexpressing *GPA1*, cells of the *crk1Δ* mutant and the *CRK1^ΔPEST^* strain expressing *P_GDP1*-GPA1:FLAG (*GPA1^OE^*) were used to represent typical cells and titan cells, respectively. We separated the membrane and cytosolic fractions of *crk1Δ GPA1^OE^* cells and *CRK1^ΔPEST^ GPA1^OE^* cells. Samples were normalized to the same dry weight and were analyzed for the presence of the Gpa1:FLAG signal in each fraction. As shown in Fig. 4l, both strains exhibit comparable Gpa1 levels in the cytosolic fraction, whereas the *CRK1^ΔPEST^ GPA1^OE^* cells exhibits more Gpa1 in the membrane fraction compared to the *crk1Δ GPA1^OE^* cells. These results suggest that Crk1-mediated Gpa1 phosphorylation leads to a change in Gpa1 localization from the cytoplasm to the plasma membrane, a process required for Gpa1 signaling activation and cell enlargement. Interestingly, low levels of phosphorylated Gpa1 and titan cell production are still detected in the absence of Crk1 (Fig. 4h), suggesting that additional protein kinases are likely involved in Gpa1 phosphorylation in *C. neoformans*.

### Crk1 shapes host immune response during infection

Our previous studies demonstrate that *fbp1Δ* is hypovirulent in vivo, and mice challenged with live or heat-killed (HK) *fbp1Δ* cells develop comparably robust Th1 responses[34,50]. Therefore, we investigated the

possible role of Crk1 in shaping fungal virulence and T cell differentiation using a murine inhalation model of systemic cryptococcosis. Mice infected with H99 or the *crk1Δ* mutant showed similar survival rates, suggesting that *crk1Δ* has no significant virulence attenuation (Supplementary Fig. 7a). In contrast, mice infected with *CRK1^OE^* or *CRK1^ΔPEST^* strains survived significantly longer, indicating attenuated virulence (Fig. 5a). The level of virulence attenuation observed in mice infected with the *CRK1^ΔPEST^* strain was more pronounced than the *CRK1^OE^* strain. We examined fungal lesion development and cell population diversity in infected lung sections by H&E staining at the end timepoint. Consistent with the in vitro and in vivo data, the lungs of mice infected with *crk1Δ* showed few observable titan cells compared to mice infected with wild type H99. In contrast, lung sections obtained from mice infected with the *CRK1^OE^* and *CRK1^ΔPEST^* strains showed an increased presence of titan cells (Fig. 5b). Based on these observations we further analyzed the virulence and immune responses elicited by *CRK1^OE^* and *CRK1^ΔPEST^* infections. Mice infected with H99 succumbed 19 days after infection as expected, whereas those infected with *CRK1^OE^* and *CRK1^ΔPEST^* strains exhibited a median survival time of 27 and 59 days, respectively (Fig. 5a). The more severe virulence attenuation observed in the *CRK1^ΔPEST^* strain is consistent with the Crk1 protein level difference in *CRK1^ΔPEST^* and *CRK1^OE^* strain (Supplementary Fig. 4). As a control, we also infected mice with the *GPA1* dominant active allele (*GPA1^Q284L^*), which is known to produce a high proportion of titan cells[21]. Our data showed that the *GPA1^Q284L^* strain also displayed significant virulence attenuation in the murine model with a median survival time of 56 days, similar to the survival rate of the *CRK1^ΔPEST^*-infected mice (Supplementary Fig. 7b). These data demonstrate that strains with overproduction of titan cells exhibit diminished fungal virulence.

Since mice infected with the *fbp1Δ* mutant developed a high Th1 protective response[34,48], we examined the cytokine profile of CD4^+ T cells recovered from the airways and lung-draining mediastinal lymph nodes (MLNs) of mice infected with *CRK1^ΔPEST^* or *CRK1^OE^*, using previously reported methods (Supplementary Fig. 7c)[34]. We observed that mice infected with the *CRK1^ΔPEST^* strain showed increased frequencies of IFN-γ^+ and IL-17A^+- producing CD4^+ T cells in the airways compared to mice infected with H99, an observation similar to what occurs in mice infected with the *fbp1Δ* mutant (Fig. 5c, d)[34,48]. We also observed increased differentiation of *Cryptococcus*-specific, IFN-γ-secreting Th1 cells and IL-17A-secreting Th17 CD4^+ T cells in the MLNs (Fig. 5e, f). Mice infected with the *CRK1^OE^* strain also showed increased CD4^+ T cells with higher Th1 cytokine production compared to mice infected with H99, but this increase did not reach statistical significance. These observations alongside our data showing that the *CRK1^ΔPEST^* strain exhibits increased Crk1 protein accumulation compared to the *CRK1^OE^* strain (Supplementary Fig. 4) suggest that protein levels of Crk1 in *Cryptococcus* influence the differentiation of CD4^+ T cells in the infected host. Taken together, our data indicate that Crk1 positively regulates titan cell production and is involved in Fbp1-mediated fungal virulence.

## Discussion

Cell size regulation is a fundamental and complex process in cell development. Many studies have investigated the impact of size control in rapidly proliferating cells (e.g., exponential growing populations of yeast cells or mammalian cancer cells). These studies reveal that cell growth and cell division are tightly coordinated to control cell size[51–54]. However, the molecular mechanisms underlying cell size control for cells undergoing non-exponential growth, such as those observed in the heterogeneity of *C. neoformans* cell populations during infection, remains elusive.

In this study, we uncover a regulatory mechanism of cell size control and fungal virulence that is centered on the stability of a CDK-related kinase Crk1. We demonstrate that Crk1 levels are regulated by

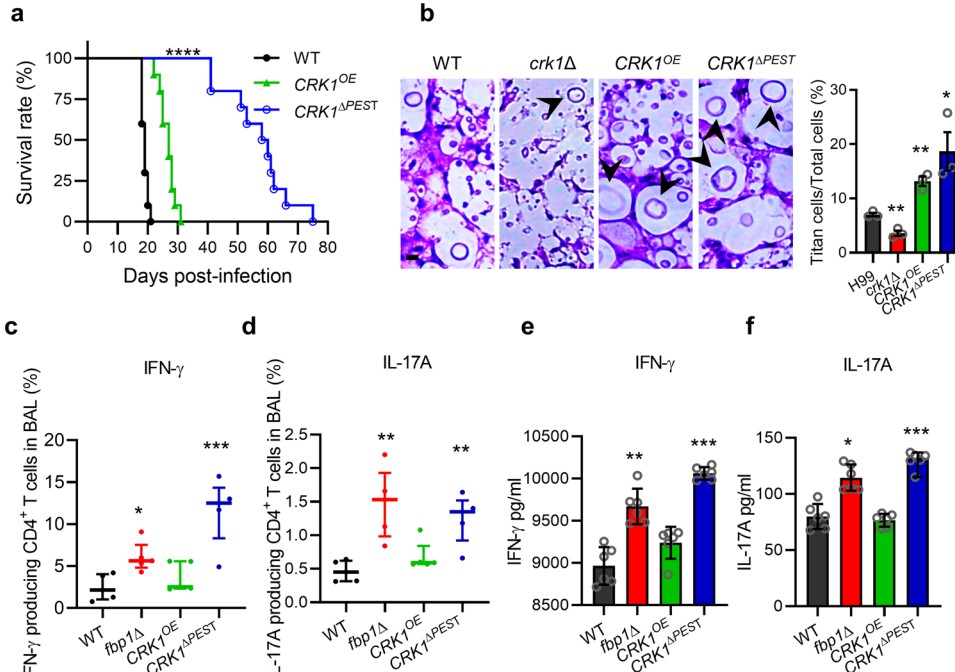

**Fig. 5 | Crk1 shapes the host immune response and fungal virulence. a** Survival curves for mice after intranasal infection with H99, *CRK1^OE*, and *CRK1^ΔPEST* strains, respectively. Statistical analysis was performed based on Log-rank (Mantel-Cox) test. ****P < 0.0001. **b** H&E-stained slides were prepared from cross sections of infected mouse lungs at endpoint and visualized by light microscopy. Bar, 10 μm. Titan cell percentage was quantitatively measured from infected lung sections at the endpoint. The titan cell percentage data are cumulative from three mice. Error bar indicates 95% confidence interval of the median. Statistical analysis was performed with the two-sided Kruskal-Wallis nonparametric test for multiple comparisons. *P = 0.03; **P = 0.004; 0.003. **c, d** Cytokine expression was analyzed by intracellular cytokine staining. Plots of cytokine production in CD4⁺ T cells was gated as Thy1.2⁺ CD4⁺ CD8⁻ T cells. The frequencies of IFN-γ- (**c**) and IL-17A- (**d**)

producing CD4⁺ T cells in BALF were analyzed. Each symbol represents one mouse. Error bar indicates 95% confidence interval of the median for 4 mice. Statistical analysis of data was done with the two-sided Kruskal-Wallis nonparametric test for multiple comparisons. *P = 0.019; **P = 0.002; 0.007; ***P = 0.0008. **e, f** *Cryptococcus*-specific CD4⁺ T cell responses were examined in lung-draining lymph nodes (MLNs) of mice after infection. Production of IFN-γ (**e**) and IL-17A (**f**) was measured by ELISA. The data shown are cumulative from three independent experiments with five mice per group. Error bar indicates 95% confidence interval of the median. Statistical analysis of data was done with the two-sided Kruskal-Wallis nonparametric test for multiple comparisons. *P = 0.02; **P = 0.004; ***P < 0.001. Source data are provided as a Source Data file.

the SCF(Fbp1) E3 ligase through its ubiquitination and degradation. Crk1-mediated Gpa1 phosphorylation stimulates Gpa1 membrane localization to activate Gpa1-cAMP signaling, which then regulates cell size in *C. neoformans*. We have demonstrated that the UPS is required for cell size regulation in *C. neoformans* and identified the role of Fbp1 in this process. Deletion of *FBP1* increases cell body size and titan cell percentage compared to its parental wild type strain H99. We also identified Crk1 as a substrate of Fbp1. We showed that Crk1 directly interacts with Fbp1 through its C-terminus and that the degradation of Crk1 is dependent on its PEST domain and Fbp1. We observed that the titan cell percentage and the median cell body size decrease in the *crk1Δ* mutant and increase in the *CRK1^OE* and *CRK1^ΔPEST* strains in both the murine model of cryptococcosis and under in vitro titan cell inducing conditions. The more extreme virulence attenuation observed in the *CRK1^ΔPEST* strain indicates a connection between titan cell production and fungal virulence, as well as an impact on host immune response.

Both Fbp1 and Crk1 are required for meiosis[55–57]. Our observation that both the *fbp1Δ* mutant and the *CRK1^OE* strain fail to undergo meiosis and sporulation prompts our hypothesis that the role of Crk1 in the meiosis process is associated with titan cell production. This is consistent with previous reports that (i) pheromone secretion by mating type α cells triggers the enlargement of mating type a cells[58], (ii) mice infected with opposite mating type strains show an increased proportion of titan cells in vivo and deletion of the pheromone receptor Ste3a in these cells decreases titan cell percentage[16] and (iii) the meiosis-specific genes *DMC1* and *REC8* contribute to *Cryptococcus* cell gigantism in vivo in a murine model of systemic cryptococcosis[20].

Similar to the *fbp1Δ* mutant, both *dmc1Δ* and *rec8Δ* mutants fail to sporulate but display normal filamentation[20,32]. It is conceivable that Crk1 modulates the development of titan cells independent of the MAPK mating pathway, because deletions of genes encoding members of the MAPK pathway and its downstream transcription factors do not affect their ability for titan cell formation[21].

We also identified a Crk1 signal when cells were grown in YPD culture conditions (Fig. 2d), indicating that Crk1 is not a meiosis-specific protein and may exhibit cellular functions other than regulating meiosis processes in *C. neoformans*, which is different from its Ime2 homolog in *S. cerevisiae*[56,59]. Indeed, we found that Crk1 is involved in the regulation of Gpa1-cAMP signaling and fungal virulence. Ime2 homologs from various fungal species also function in diverse cellular processes[60]. ImeB in *Aspergillus nidulans* is required for mycotoxin production[61]. Ime2 in *Neurospora crassa* negatively regulates protoperithecia formation, non-self recognition, and cell death[62,63]. Ime2 in *Arthrobotrys oligospora* plays roles in mycelial growth, conidiation, osmoregulation, and pathogenicity[64].

The Gpa1-cAMP signaling is critical in fungal virulence and regulates titan cell formation[16,17,21,22]. Here we found that Crk1 interacts with Gpa1 and functions upstream of Gpa1-cAMP signaling based on epistatic genetic analysis. Our findings further indicate that Crk1 likely regulates Gpa1 function through phosphorylation under titan cell inducing conditions. Both the Crk1 overaccumulating strain (*CRK1^ΔPEST*) and the *GPA1* dominant active allele (*GPA1^Q284L*) produce a high proportion of titan cells and showed significant virulence attenuation in a murine infection model, which is consistent with the role of Crk1 in Gpa1 phosphorylation and activation. Phosphorylation of the G

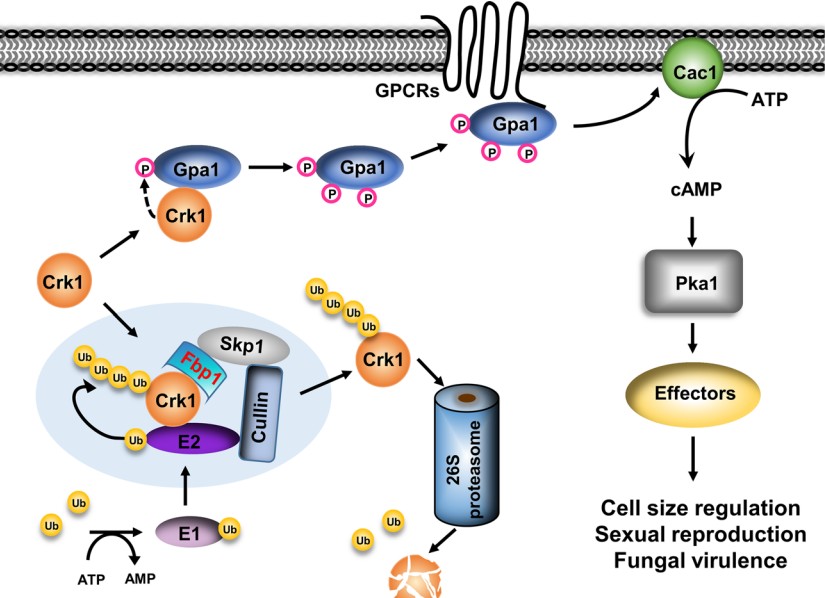

**Fig. 6 | Proposed model for the involvement of Crk1 in the ubiquitin pathway and the G protein pathway.** The SCF(Fbp1) E3 ligase interacts with Crk1 for its ubiquitination and degradation at the 26 S proteasome. Overproduction of Crk1 under titan cell-inducing conditions promotes Gpa1 phosphorylation. The phosphorylated Gpa1 is translocated to the plasma membrane for its binding with the G protein-coupled receptor, which activates Gpa1-cAMP signaling to regulate cell enlargement.

protein alpha subunit as a regulatory mechanism in G protein signaling was first observed in the early 1990s[65,66]. In *S. cerevisiae*, global phosphoproteomics studies identify Gpa2 as a potential phosphoprotein[67]. Gpa2 undergoes nutrient-regulated phosphorylation, which enhances plasma membrane localization of Gpa2 and promotes protein kinase A activation[47]. Modification of Ime2 in *S. cerevisiae* is required for the phosphorylation of Gpa2, suggesting a feedback regulation of Ime2 on Gpa2[68]. Our data are consistent with these findings that Gpa1 can be phosphorylated to activate PKA signaling. It should be noted that Gpa1 phosphorylation by Crk1 was detected under titan cell inducing conditions, but not in the other conditions we tested, thereby indicating a specific role of Crk1 in the regulation of titan cell production. The Gpa1-cAMP pathway in *C. neoformans* has been well studied, however the Gpa1 phosphorylation during titan cell regulation has not been reported before. Our identification of Crk1 as a CDK-related kinase in Gpa1 post-translational modification may lead to a better understanding of this key signaling pathway.

Although the modest increase in Th1 and Th17 cytokine production in CD4+ T cells in response to the *CRK1^OE* strain infection did not reach statistical significance when compared to H99 infection, we detected a significantly increased induction of Th1 and Th17 CD4+ T responses in *CRK1^ΔPEST*-infected mice, which indicates that Crk1 protein level is important for regulation of fungal virulence[33]. Virulence attenuation of *CRK1^ΔPEST*, *fbp1Δ*, and *GPA1^Q284L* strains is paradoxical to their overproduction of titan cells because titan cells are known to contribute to cryptococcal virulence[16,17,21,69,70]. However, these observations are consistent with reports on *USV101* and *PDR802* genes, which negatively regulate titan cell formation[23,24,26,71]. The *usv101Δ* mutant impairs crossing in an in vitro blood-brain barrier model, and delays dissemination to the brain in the murine infection model[71]. These hypovirulent mutants produce more titan cells, suggesting that fungal virulence is a complex trait governed by multiple factors (e.g., cell size). It is possible that the overproduction of titan cells limits or delays yeast dissemination during infection, leading to virulence attenuation. These observations demonstrate that precise regulation of virulence factors and protein levels is critical for cryptococcal pathogenicity. Additionally, virulence attenuation and other defects in

cell function due to improper protein levels have been documented[72]. The titan cells produced in wild type H99 reportedly induce a Th2 response and promote infection[70,73,74], which differs from our data showing that increased titan cells in our Crk1 overproducing strains induced a higher Th1 response. The mechanisms of the host immune activation during *Cryptococcus* infection are likely complex. Despite a higher frequency of large cells, it is possible that our mutant strains may have other cell surface structural changes that in turn promote robust Th1 and Th17 CD4+ T cell responses. Indeed, the studies on Rim101, a downstream transcription factor of the cAMP/PKA pathway, showed that deletion of *RIM101* alters cell surface structure and blocks titan cell production[75,76]. The cell surface component chitin both stimulate and inhibit immune responses[74,75]. The *rim101Δ* mutant helps to expose cell surface antigens to the immune system and increase proinflammatory cytokine levels, including strong Th1 and Th17 responses[75]. The *fbp1Δ* mutant and the *CRK1^ΔPEST* strain may have similar cell surface changes as the *rim101Δ* mutant, and these possibilities need to be investigated in future studies.

In summary, our study reveals a regulatory mechanism of cell size control in *C. neoformans* that involves UPS and its regulation of a CDK-related kinase. Based on our data, we propose a model whereby Crk1 interacts with both Fbp1 and Gpa1 to regulate meiosis and titan cell production in *C. neoformans* (Fig. 6). In this model, Crk1 interacts with Fbp1 through the C-terminus of Crk1 and functions as the downstream target of Fbp1. In the *fbp1Δ* mutant, Crk1 protein accumulates and regulates Gpa1 phosphorylation to activate the cAMP/PKA signaling pathway, thereby inducing titan cell formation. It is possible that additional factors are involved in this Crk1 and Fbp1-regulated mating process and cell size change. Our finding that Crk1 regulates Gpa1 phosphorylation to activate the Gpa1-cAMP signaling pathway reveals a new level of Gpa1 G protein signaling regulation. The identification of Crk1 as a substrate of the SCF E3 ligase connects the UPS with G protein signaling during titan cell induction to control cell size change, sexual reproduction, and fungal virulence, consequently providing insight into cell size regulation. The observation that there is a conserved role for meiosis machinery in the regulation of cell size in humans, model yeast and *C. neoformans* may suggest an evolutionarily conserved

mechanism. Further studies in this area will provide new mechanistic insight on the link between morphogenesis and pathogenesis in other clinically important fungal pathogens.

## Methods

### Ethics statement

All animal studies were conducted following biosafety level 2 (BSL-2) protocols and procedures approved by the Institutional Animal Care and Use Committee (IACUC) and Institutional Biosafety Committee of Rutgers University, respectively. The studies were conducted in facilities accredited by the Association for Assessment and Accreditation of Laboratory Animal Care (AAALAC). Mice were housed in groups of five in individually ventilated cages at $21 \pm 1\,°C$, 30–70% relative humidity, 12 h/12 h dark/light cycle from 7:00 am–7:00 pm, with free access to food and water, and autoclavable mouse houses as environmental enrichment.

**Strains and media.** Strains and plasmids used in this study are listed in Supplementary Table 1. All primers used in this study are listed in Supplementary Table 2. YPD medium (2% peptone, 1% yeast extract, and 2% glucose) was used for the routine culture of *C. neoformans* strains. A V8 juice agar medium (50 mL V8 original juice, 0.05% $KH_2PO_4$, and 2% agar, pH 5.0) and a modified Murashige and Skoog (MS) medium minus sucrose (Sigma–Aldrich, Steinhelm, Germany) were used for mating and sporulation assays[77]. Minimal medium (MM, 15 mM D-glucose, 10 mM $MgSO_4$, 29.4 mM $KH_2PO_4$, 13 mM Glycine, and 3.0 μM Thiamine. pH5.5) was used for titan cell induction[24,78]. Strains using the *CTR4-2* inducible promoter were grown in YPD media containing 25 μM $CuSO_4$ and 1 mM ascorbic acid to suppress the expression of the examined genes, or 200 μM copper chelator bathocuproinedisulphonic acid (BCS) to drive gene expression[79].

**Titan cell induction and measurement in vitro.** Titan cell inductions were performed as previously reported[24]. Strains were grown overnight at 30 °C in YPD liquid medium. Cells were collected, washed twice with minimal medium, and suspended in minimal medium with a final concentration of $10^6$ cells/mL. One mL of suspension was incubated at 30 °C for 3 days in a 1.5 mL Eppendorf tube with cap closed. Body sizes of more than 100 cells were measured and the percentage of titan cells (Cell body size over 10 μm) was calculated.

**DNA content measurement.** DNA content was measured as described previously[20]. Briefly, cells were fixed in ice-cold 70% ethanol overnight at 4 °C after 3 days of incubation in titan cell inducing conditions. The fixed cells were washed, resuspended in 0.5 ml of NS buffer (10 mM Tris-HCl, 0.25 M sucrose, 1 mM EDTA, 1 mM $MgCl_2$, 0.1 mM $ZnCl_2$, 0.4 mM phenylmethylsulfonyl fluoride, and 7 mM β-mercaptoethanol). RNase A (0.5 mg/ml) and propidium iodide (10 μg/ml) were added into the suspension and incubated for 2 h at 37 °C in the dark. The cells were sonicated for 10 s before analysis with BD FACSVia™ flow cytometer (BD Biosciences) with BD FACSDiva software v8.0. Data were analyzed with FlowJo software v10.

**DAPI staining.** DAPI (4,6-diamidino-2-phenylindole) staining was performed as previously reported[32]. Briefly, Cell cultures and mating filaments were fixed with formaldehyde (9.3%) for 10 min. The fixed cells were washed twice with PBS, permeabilized with an equal volume of PBS buffer containing 1% Triton X-100 for 5 min, washed twice again with PBS, and resuspended in PBS. Equal volumes of cell suspension and DAPI mixture (20 ng/ml DAPI, 1 mg/ml antifade, and 40% glycerol) were mixed and observed with a Nikon fluorescence microscope.

**Mating assay.** *C. neoformans* cells of opposite mating types were mixed and cultured on V8 or MS agar medium at 25 °C in the dark. Mating filaments and basidiospores formation were examined and photographed following 7 or 14 days incubation.

**Protein–protein interaction assay using the yeast two-hybrid system.** Yeast two-hybrid interaction assays were performed as previously described[43]. Briefly, the *FBP1* or *GPA1* full-length cDNA were cloned into the bait vector pGBKT7 and fused with the BD domain. cDNA of *CRK1* and C-terminal region of *CRK1* (*CRK1CT*) were cloned into the prey vector pGADT7 and fused with the AD domain. All inserted cDNA sequences were confirmed through sequencing. Both bait constructs and prey constructs were co-transformed into the yeast strain PJ69-4A. Transformants growing on SD medium lacking histidine or adenine were considered as positive interactions. The expression of the LacZ gene in these transformations was quantified by β-galactosidase enzyme activity assays using chlorophenol red-β-D-galactopyranoside (CPRG) (Calbiochem, San Diego, CA) as a substrate.

**Protein–protein interaction using co-immunoprecipitation assay.** The *FBP1* full-length cDNA was amplified with primers CX225/CX443 (Supplementary Table 2). The *FBP1* cDNA lacking the F-box domain was amplified by overlap PCR using primers CX225/CX198 and CX199/CX443. Both fragments were cloned into the BamHI/NotI sites of a vector containing the *Cryptococcus* actin promoter[80] and a FLAG epitope, to generate plasmids pCXU118 and pCXU117 which contain *FBP1:FLAG* and *FBP1^{ΔF}:FLAG* fusions, respectively. Linearized pCXU118 or pCXU117 were introduced into CUX87 (*fbp1Δ ura5Δ*) to generate strains CUX138 and CUX135 that express Fbp1:FLAG and Fbp1^{ΔF}:FLAG proteins, respectively. The *CRK1* cDNA was amplified with primers CX394/CX395 and cloned into the BamHI sites of pCTR4-2 vector[79] to generate the *CRK1:HA* fusion plasmid pCXU108. pCXU108 was biolistically transformed into CUX138, CUX135, and CUX87 to generate strains CUX140 (Fbp1:FLAG Crk1:HA), CUX141 (Fbp1:FLAG Crk1:HA), and CUX119 (Crk1:HA), respectively. Total proteins were purified and analyzed by immunoblotting with anti-FLAG and anti-HA antibodies. Proteins were pulled down by using anti-FLAG affinity gel (Sigma, A2220) and analyzed by immunoblotting with anti-FLAG (1:2000) to detect Fbp1:FLAG and Fbp1^{ΔF}:FLAG or with anti-HA antibodies (1:2000) to detect Crk1:HA. Unprocessed scans of the blots are provided in the Source Data file.

**Detection of protein accumulation and stability.** A Crk1:HA fusion construct, in which *CRK1* was fused with an HA tag at its C-terminus and was controlled by its native promoter, was transformed into H99 and the *fbp1Δ* mutant background to generate strains CUX222 ($P_{CRK1}$-*CRK1:HA*) and CUX221 (*fbp1Δ* $P_{CRK1}$-*CRK1:HA*). Cells were harvested after being cultured in YPD overnight and the abundance of the Crk1:HA protein was measured by western blotting. To test the stability of Crk1 and Crk1^{ΔPEST} in H99 and *fbp1Δ* mutant strain backgrounds, Crk1:HA- tagged strains CUX118 and CUX119 and Crk1^{ΔPEST}:HA-tagged strains CUX1131 and CUX1132 were first grown in YPD medium with 200 μM BCS to induce the *CTR4* promoter and then washed with PBS. Washed cultures were transferred to YPD containing 25 μM $CuSO_4$ and 1 mM ascorbic acid to block the transcription of *CRK1:HA*. Cells were collected after 0, 1, 2, 4, and 6 h of incubation, and protein extracts were prepared as previously described[80]. Crk1:HA and Crk1^{ΔPEST}:HA were detected by western blotting using a monoclonal anti-HA antibody (Genscript, A01244, 1:2000).

### Quantitative RT-PCR

The wild type H99, the *CRK1^{OE}* strain, and the *CRK1^{ΔPEST}* strain were cultured overnight under YPD condition to examine the relative expression levels of *CRK1*. The wild type H99, the *crk1Δ* mutant, the *CRK1^{OE}* strain, and the *fbp1Δ* mutant were under titan cell inducing conditions to examine the relative expression levels of *PKA1*. Total RNAs were extracted using Trizol reagent (Invitrogen) and purified with the Qiagen RNeasy cleanup kit (Qiagen) following the manufacturer's instructions. First-strand cDNAs were synthesized using a Superscript III cDNA synthesis kit (Invitrogen) following the

manufacturer's instructions[32]. Genes expression was analyzed using SYBR advantage QPCR premix reagents (Clontech). Real-time PCRs were performed using an Agilent AriaMx Real-Time PCR System with the AriaMx Software v1.7. Gene expression levels were normalized using the endogenous control gene *GAPDH*, and relative levels were determined using the comparative threshold cycle (CT) method[81].

### RNA-seq quantification analysis

*C. neoformans* cells of the WT, *fbp1*Δ, *CRK1^OE^*, and *CRK1^ΔPEST^* strains were cultured overnight at 30 °C in YPD liquid medium and then inoculated in titan cell inducing conditions for 3 days. The cells were then collected and total RNA was extracted following the protocol as described previously[32]. RNA-Seq analysis was performed by Novogene according to the company's protocol (https://en.novogene.com). The library preparations were sequenced on an Illumina platform and paired-end reads were generated. Approximately 20 million (M) raw reads were obtained by sequencing each library. The raw reads were filtered by removing the adapter and low quality reads. Clean reads were mapped to the annotated genome of *C. neoformans* H99 using HISAT2 software (version 2.0.5). Differential expression analyses were conducted using the DESeq2 package (version 1.20.0)[82]. To be considered significantly differentially- expressed, a gene must meet three criteria: (1) a FPKM value greater than or at least equal to 20 in one sample; (2) an absolute $\log_2$(fold change) value greater than or equal to 1.0; and (3) a *P* value [adjusted for the false discovery rate (FDR)[82]] lower than 0.01.

### Assays for cAMP production

cAMP assays were conducted as described previously[42]. Briefly, a single colony of each *C. neoformans* strain was inoculated into 10 ml of YPD liquid medium and incubated for 24 h at 30 °C with shaking. Cells were collected and washed twice with ddH₂O, once with MES buffer (10 mM MES, 0.5 mM EDTA), and resuspended in 20 ml MES buffer. 15 μl of diluted cells (OD₆₀₀ = 2.0) were incubated for 2 h at 30 °C for glucose starvation. 1 ml of cells was filtered through a wet Millipore filter on a vacuum manifold (pore size 0.45 micron, HVLP02500) for the 0 time point. 1.5 ml of 20% glucose was added to the remaining 14 ml of cell suspension. 1 ml was removed and filtered at 30 s, 1 min, and 3 min. At each time point, filters were immediately removed, placed into Petri-dishes containing 1 ml of formic acid solution (9.2 ml of 100% formic acid, 190.8 ml dH₂O, and 50 ml butanol), and agitated for 1 h to lyse cells on a table-top orbital shaker. Cell suspensions were spun down and the supernatants were transferred to fresh tubes and lyophilized. Pellets were resuspended in 400 μl of assay buffer and 100 μl was used for each sample. cAMP concentrations were determined using the cAMP Biotrak Enzyme immunoassay (EIA) system (Amersham, Piscataway NJ) and normalized to $10^6$ cells.

### In vivo ubiquitination assays

H99, CUX1340 (P_CTR4_-HA), CUX118 (P_CTR4_-Crk1:HA) and CUX119 (*fbp1*Δ P_CTR4_-Crk1:HA) were cultured in YPD overnight. Cells were collected, washed and cultured in YPD in the presence or absence of 20 μM MG132 for 4 h. Total proteins were extracted in lysis buffer (50 mM Tris-HCl, 150 mM NaCl, 1% Triton X-100, 1 mM EDTA, 10 μM PMSF and 1×EDTA-free protease inhibitor) and Crk1:HA was purified by HA antibody immunoprecipitation (Genscript, A01244). Accumulation of Crk1 polyubiquitination (Crk1-(Ub)n) was detected using anti-HA antibody (Genscript, A01963, 1:2000).

### Phosphatase assays

CUX1196 (P_GDP1_-GPA1:FLAG) was cultured in titan cell inducing conditions for 3 days. Cells were harvested and stored at −80 °C. Samples were lyophilized and the mass of dried samples were measured. Lyophilized samples were subsequently pulverized with a ceramic mortar and a pestle and resuspended in phosphatase buffer (100 mM NaCl, 50 mM Tris-HCl, 10 mM MgCl₂, 1 mM dithiothreitol, and 1× EDTA-free protease inhibitors). Each cell resuspension was split in

half and subjected to lysis in the absence or presence of phosphatase inhibitors (Sigma, P5726). Lysates were centrifuged and the supernatant fraction was then collected. Total protein extracts were treated with or without protein phosphatase (Sigma, P7640) for 30 min at 30 °C. The reaction was stopped by adding SDS-PAGE loading buffer and analyzed by immunoblotting with FLAG antibody (Genscript, A01868, 1:2000). Alternatively, phosphatase assays were conducted on purified Gpa1 protein. Pulverized samples were resuspended in lysis buffer (50 mM Tris-HCl, 150 mM NaCl, 1% Triton-100, 1 mM EDTA, 10 μM PMSF and 1× EDTA-free protease inhibitors), and the cleared whole cell lysate was incubated with anti-FLAG M2 affinity gel (Sigma, A2220) for 2 h. After 3 times of washing with TBS buffer (50 mM Tris-HCl, 100 mM NaCl), the resin was incubated with 3xFLAG peptide (Sigma, F4799) at 4 °C for 30 min to elute the purified Gpa1:FLAG. Purified protein was treated with or without protein phosphatase for 30 min at 30 °C. The reaction was stopped by adding SDS-PAGE loading buffer and analyzed by immunoblotting.

**In vitro kinase assay.** CUX1196 (P_GDP1_-GPA1:FLAG) strain was cultured overnight in YPD. Cells were lysed in phosphatase buffer. The cleared whole-cell lysate was treated with protein phosphatase for 30 min at 30 °C and then incubated with anti-FLAG M2 affinity resin. The Gpa1:FLAG was eluted by 3xFLAG peptide. CUX118 (P_CTR4_-CRK1:HA) was cultured in titan cell inducing condition for 3 days. Cells were lysed, and the cleared whole-cell lysate was incubated with HA antibody (Genscript, A01244, 1:2000) and protein A agarose beads (Santa Cruz Biotechnology, sc-2001). After washing with kinase buffer (9802, Cell Signaling Technology) for 3 times, an aliquot of purified Gpa1:FLAG was added to the beads, and ATP was added (or not) to initiate the kinase reaction. SDS-PAGE loading buffer was added to stop the reaction after 2 h treatment at 30 °C. The kinase reaction was analyzed by immunoblotting with FLAG antibody (Genscript, A01868, 1:2000).

**Gpa1 phosphorylation.** A Gpa1:FLAG fusion construct was transformed into H99, the *crk1*Δ, the *crk1*Δ+*CRK1*, the *CRK1^OE^*, the *CRK1^ΔPEST^*, and the *fbp1*Δ backgrounds to generated strains CUX1196 (Gpa1:FLAG), CUX1197 (*crk1*Δ Gpa1:FLAG), CUX1200 (*crk1*Δ+*CRK1* Gpa1:FLAG), CUX1198 (*CRK1^OE^* Gpa1:FLAG), CUX1331 (*CRK1^ΔPEST^* Gpa1:FLAG), and CUX1342 (*fbp1*Δ Gpa1:FLAG). Strains were cultured in titan cell inducing conditions and cell pellets were resuspended in lysis buffer in the presence of phosphatase inhibitors (50 mM Tris pH 7.5, 1% sodium deoxycholate, 5 mM sodium pyrophosphate, 50 mM NaF, 0.1% SDS, 1% Triton X-100, 10 μM PMSF, 1x protease inhibitor and phosphatase inhibitor). Total proteins were extracted and analyzed by western blotting with FLAG antibody (Genscript, A01868, 1:2000).

**Gpa1 localization.** Membrane and cytosolic proteins were isolated following published protocols with modifications[83,84]. Briefly, the same dry weight of titan cells (CUX1331, *CRK1^ΔPEST^* P_GDP1_-GPA1:FLAG) and typical cells (CUX1197, *crk1*Δ P_GDP1_-GPA1:FLAG) were used for protein extraction. CUX1331 and CUX1197 were harvested and stored at −80 °C after culturing in titan cell inducing condition for 3 days. Samples were lyophilized and the same dry weight of lyophilized samples were subsequently pulverized with a ceramic mortar and a pestle and resuspended in 1.5 ml lysis buffer (50 mM Tris-HCl, 150 mM NaCl, 1 mM EDTA, 10 μM PMSF and 1× EDTA-free protease inhibitor). Cells were centrifuged at $2000 \times g$ for 10 min at 4 °C to remove non-lysed cells from total cells lysates. The supernatant was transferred to a new tube and separated into a soluble and pellet fraction by centrifugation at $25,000 \times g$ for 60 min at 4 °C. The insoluble pellet was resuspended in 0.1 ml lysis buffer with 1% Triton X-100 to extract membrane proteins. 10 μl of soluble cytosolic proteins and 5 μl of membrane proteins were analyzed by western blotting with FLAG antibody (Genscript, A01868, 1:2000) and actin antibody (Genscript, A00702, 1:5000), marker of cytoplasm.

**Fluorescence imaging.** A construct expressing Gpa1:GFP fusion protein with the ACT promoter was transformed into H99, the $crk1\Delta$, and the $crk1\Delta+CRK1$ strain backgrounds to generate strains CUX1235 (Gpa1:GFP), CUX1236 ($crk1\Delta$ Gpa1:GFP), and CUX1237 ($crk1\Delta+CRK1$ Gpa1:GFP). Strains were cultured overnight and transferred to YPD medium or titan cell inducing conditions. Cells were observed with a Nikon laser confocal fluorescence microscope after 3 days of incubation in titan cell inducing conditions or overnight culture in YPD conditions. Constructs expressing $P_{ACT1}$-CRK1:mCherry fusion protein (pCXU401) or $P_{ACT1}$-CRK1$^{\Delta PEST}$:mCherry fusion protein (pCUX402) were linearized by XhoI and transformed into H99 to generate the $CRK1^{OE}$ and $CRK1^{\Delta PEST}$ strains. Strains were cultured in YPD overnight and were observed with a Nikon laser confocal fluorescence microscope.

**Virulence studies.** Yeast strains were grown at 30 °C overnight and cultures were washed twice with PBS buffer, and resuspended to a final concentration of $1 \times 10^6$ cells/mL. Groups of 10 female BALB/c or C57BL/6 mice were intranasally infected with $5 \times 10^4$ yeast cells of each strain as previously described[85]. Over the course of the experiments, animals that appeared moribund or in pain were sacrificed by $CO_2$ inhalation. Survival data from the murine experiments were statistically analyzed between paired groups using the long-rank test (P values of < 0.05 were considered statistically significant). Besides animal survival, a subgroup of infected lungs was harvested at 3-, 7-, and 15-days post-inoculation and the ETP (end time point for H99 and $fbp1$ complement strain or 50 days for $fbp1\Delta$ mutant). Fixed lungs were stained with Hematoxylin and Eosin (H&E) and examined by light microscopy. Body sizes of more than 50 cells were measured and cell populations were calculated.

**Histopathology and fungal burden in infected organs.** Infected animals were sacrificed at designated time points and at the endpoint of the experiment according to the Rutgers IACUC-approved animal protocol. Infected lungs and brains were isolated, fixed in 10% formalin solution, and sent to the Rutgers Histology Core facility. Tissue slides were stained with Hematoxylin and Eosin (H&E) and examined by light microscopy. Infected lungs, brains, and spleens were isolated and homogenized using a homogenizer in $1\times$ PBS buffer. Resuspensions were diluted and 100 μl of each dilution was spread onto YPD medium with ampicillin and chloramphenicol and colonies were identified following 3 days of incubation.

**Isolation and measurement of yeast cells from mouse BALF.** Yeast strains were grown at 30 °C overnight and cultures were washed twice with PBS buffer, and resuspended to a final concentration of $1 \times 10^8$ cells/mL. Groups of 4 female C57BL/6 mice were intranasally infected with $5 \times 10^6$ yeast cells of each strain as previously described[85]. BALF samples were harvested at day 3 after inoculation. BALF was collected in 3 ml of $1\times$ PBS buffer using a catheter inserted into the trachea of animal post-euthanasia, and airway-infiltrating cells were lavaged with ~0.7 ml of $1\times$ PBS at one time to a total volume of 3 ml. Cells were spun down and host cells were lysed by adding sterile $ddH_2O$ and incubating for 45 min at room temperature. *Cryptococcal* cells were examined under microscope. Body sizes of more than 500 cells were measured and the titan cell (Cell body size over 10 μm) proportion was calculated.

**Intracellular cytokine staining of T cells harvested in BALF and flow cytometry.** Yeast strains were grown at 30 °C overnight and cultures were washed twice with PBS buffer, and resuspended to a final concentration of $2 \times 10^7$ cells/mL. Groups of 5 female C57BL/6 mice were intranasally infected with $1 \times 10^6$ yeast cells of each strain. BALF samples were harvested at day 7 after infection as previously described[34,50]. All collected cells were pelleted and resuspended in 200 μl of RPMI containing 10% fetal calf serum (FCS), penicillin-

streptomycin (2200 U/ml; Gibco), and a gentamicin sulfate solution (1 mg/ml). BALF cells were then plated in a 96-well tissue culture treated plate and restimulated using BD-leukocyte activation cocktail containing BD GolgiPlug (BD Biosciences) according to the manufacturer's instructions. Four hours after activation, BALF cells were surface stained with fluorescently labeled antibodies against Thy1.2, CD4, and CD8. Samples were fixed in 1% paraformaldehyde overnight. Prior to intracellular staining, the samples were permeabilized with $1\times$ BD Perm/Wash buffer according to the manufacturer's instructions. Intracellular cytokine staining (ICCS) was done using fluorescently labeled antibodies against IFN-γ, IL-17A diluted in $1\times$ BD Perm/Wash for 45 min on ice. Samples were immediately washed and analyzed by flow cytometry as described below. BALFs were cell surface stained for T cells with Thy1.2 (53-2.1, PE-Cy7, BD Biosciences, 561642, 1:200), CD4 (RM4-5 Pacific Blue, BD Biosciences, 558107, 1:200), CD8 (53-6.7, PerCp Cy5.5, BD Biosciences, 561109, 1:200) and ICCS for IFN-γ (XMG1.2, PE, BD Biosciences, 554412, 1:100) and IL-17A (eBio17B7, APC, eBioscience, Inc., 17-7177-81, 1:100) by following standard procedures. IL-17A was obtained from eBioscience, Inc., most other antibodies and reagents for cell surface and ICCS were from BD Biosciences.

**CD4$^+$ T cell isolation and CD4 T cell recall response.** Antigen-presenting cells (APCs) were prepared from the spleen of syngeneic, uninfected donor mice. Splenic cell suspensions were depleted of T cells by antibody complement-mediated lysis. Splenic cells were incubated with anti-Thy1.2 antibodies and rabbit complement (Low Tox; Cedarlane Labs, Hornby, ON, Canada) at 37 °C for 1 h. Lung-draining lymph nodes (MLNs) were collected and placed in 10 ml pf PBS. Total lymphocyte cell suspensions were prepared by gently releasing the cells into PBS by applying pressure to the lymph nodes with the frosted ends of two glass slides. For CD4$^+$ T cell isolations, individual samples from each group were pooled (5 mice). CD4$^+$ T cells were purified using a negative-sorting CD4$^+$ isolation kit (Miltenyi Biotec, Inc., Auburn, CA). CD4$^+$ T cell isolation was done by following the manufacturer's instructions and were consistently found to be >90% pure, as assessed by flow cytometry. Purified CD4$^+$ T cells (200,000 cells) were cultured with T cell-depleted APCs (300,000 cells) in RPMI containing 10% fetal calf serum (FCS), penicillin-streptomycin (2,200U/ml; Gibco) and gentamicin sulfate solution (1 mg/ml). The cultures were plated in 96 well-plates and incubated at 37 °C with 5% $CO_2$ for 72 h. To measure *Cryptococcus*-specific CD4$^+$ T cell response, CD4-antigen-presenting cell cultures were incubated with sonicated H99 yeasts as a source of fungal antigens. The amount of antigen used was adjusted to a multiplicity of infection of 1:1.5 (antigen-presenting cell:yeast). The fungal growth inhibitor voriconazole was used at a final concentration of 0.5 mg/ml to prevent any fungal cell outgrowth during the culture period. After 72 h of culture, supernatants were collected for cytokine analysis by ELISA (IFNγ, IL-17A Invitrogen) according to the manufacturers' instructions.

**Statistics and reproducibility.** Unless otherwise specified, all data presented in the figures are representative of at least three independent experiments with similar results. Microscopical images presented in the figures are representative images of three independent experiments. Methods for statistical analysis are specified in the legend of each individual figure.

### Reporting summary
Further information on research design is available in the Nature Research Reporting Summary linked to this article.

## Data availability
All raw RNA-seq data were deposited at NCBI Sequence Read Archive (SRA) (https://www.ncbi.nlm.nih.gov/sra) with the accession number

PRJNA816899. Processed signature data can be accessed in Supplementary Data 1. The ePESTFind program is available at http://emboss.bioinformatics.nl/cgi-bin/emboss/epestfind. The STRING (Search Tool for Retrieval of Interacting Genes/Proteins) database can be found at http://string-db.org. Source data are provided with this paper.

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

## Acknowledgements

We thank Drs. Kirsten Nielsen and Xiaorong Lin for valuable discussions and comments on this study and Life Science Editors for careful editing and discussion. We thank Dr. Katsunori Sugimoto for his advice and support on the Gpa1 phosphorylation assay. The CTR4 promoter construct was kindly provided by Dr. Tamara Doering. This study is supported by NIH grant R01AI141368 to A.R. and C.X., and the Rutgers HealthAdvance Fund (partially supported through NIH U01HL150852) to C.X. Studies in the Xue lab are also supported by NIH grants R01AI123315 and R21AI154318. A.R. holds an Investigators in the Pathogenesis of Infectious Disease Award from the Burroughs Wellcome Fund.

## Author contributions

C.C., A.R. and C.X. developed hypothesis and study design. C.C. and C.X. wrote the first and successive drafts of the manuscript. C.C., K.W., A.R. and C.X. contributed to interpretation of data, and critical revision of the manuscript. C.C., K.W., Y.W., and T.L. performed experiments. A.R. and C.X. obtained funding.

## Competing interests

The authors declare no competing interests.

## Additional information

**Correspondence and requests** for materials should be addressed to Chaoyang Xue.

