## [Peer Review File · Nature Communications]

Ubiquitin proteolysis of a CDK-related kinase regulates titan cell formation and virulence in the fungal pathogen *Cryptococcus neoformans*Reviewer #1 (Remarks to the Author) [please see also the attached document]:

The manuscript by Cao and colleagues describes the function of the E3 ligase Fbp1 in titan cell formation and how its ubiquitination of the CDK-related kinase Crk1 results in control of Gpa1 and cAMP/PKA signaling that ultimately leads to cell size modifications and titan cell formation in the human fungal pathogen *Cryptococcus neoformans* that impact virulence. These studies provide improved clarity on critical unknown aspects of the signal transduction pathway involved in cell size regulation that link cell division and meiosis events in a context outside of classical mating/sporulation – a process that is not well understood in any organism but that is of critical importance, especially in the context of human cancer, adaptation, evolution, and even drug resistance mechanisms.

The studies presented are clear and well described with only minor experimental areas where the authors have overstated their conclusions or where the data are not thoroughly supported by their data. Detailed comments are listed below.

1. Example H&E sections that were analyzed to generate the data presented in Fig 1a need to be included so the reader can determine whether the data presented in accurately represent the data that are visible in the microscopic sections. Because these microscopic slides are typically only 5um tissue sections and *Cryptococcus* cells are anywhere from 2-100um in diameter, using slides for measurements of *Cryptococcus* size has a high degree of inaccuracy because it is unknown if the measurement is being taken through the middle of the cell. If serial sections are not analyzed to verify that the same cell is not analyzed on multiple slides, and that all cells have an equal chance of being represented in sections, then an indication of cell size can be gleaned from analysis of H&E sections. However, visual comparison of the slides and the data should always be presented simultaneously.

2. This reviewer has concerns with the way the statistics are presented in Figure 1. As presented, statistical significance was only observed between the wt and fbp1 mutant and not between the mutant and the complement, even though the complement appears to have a similar phenotype to the wt. If this is true, these data are very concerning and raises the question as to how statistically significance was obtained for the mutant, but not the complement in comparison to the wt, and why a difference between the mutant and the complement was also not observed. These statistics do not make any sense to this reviewer.

3. Fig 1f – indicate the difference between the black and red lines in the flow plot (assume the black line is normal conditions and the red line is in BAL but this is not stated in the figure legend or on the diagram).

4. Lines 177-180. This section needs to be rewritten as the authors step over the line between summarizing their results and making conclusions. While they can state that their data suggest that Crk1 plays important role in Fbp1-mediated fungal meiosis and sporulation. They provide no direct data in this section to link Crk1 to vegetative growth or titan cell formation. Thus, their last sentence is speculation. It would be better to lay the foundation for this argument in the previous paragraph and then more fully develop this concept in the Discussion section.

5. Fig s2b should be moved to Fig3 as this information is critical to show that the large cells generated by the CRK1OE strain are bona fide titan cells.

6. Fig 3h. "Categories" is vague. Are these GO terms? Metabolic categories?

7. The crk1Δ GPA1Q284L mutant grown under in vitro titan inducing conditions that produces enlarged cells needs to be analyzed for ploidy to confirm that the cells are bona fide titan cells and the data presented in the figure. Similarly, all of the strains in Fig 4 that are putatively producing titan cells should be analyzed for ploidy to confirm titan cell formation in this analysis and the data presented in the figure.

8. The localization data for Gpa1 is unconvincing. It all hinges on fluorescence data presenting in Figure 4i. Yet it is unclear how the authors accounted for the difference in volume/size of the cells in their determination of the fluorescence and localization of the GPA1:GFP signal. Could the decrease in cytosolic signal in the titan cell simply be due to the larger size of the titan cell resulting in the protein being distributed across a larger volume (i.e. a bigger sphere). What happens if the authors take into account the volume of the cell when measuring the internal/cytosolic fluorescence? In the Discussion section, Lines 348-350, the authors make the bold statement that "Crk1-mediated Gpa1 phosphorylation induces Gpa1 membrane localization to trigger GPCR activation of the Gpa1-cAMP signaling to regulate cell size in *C. neoformans*". In this reviewer's opinion, the authors cannot make this statement without significant additional proof of differences in Gpa1 protein localization differences in the Crk1 mutants. For example, the authors could perform sub-cellular localizations with Gpa1-FLAG strain in the different backgrounds under the titan inducing conditions and examine the membrane vs. cytosolic fractions to quantify protein levels in the different fractions.

9. Line 321 – overproduction of titan cell has previously been shown to result in decreased virulence. See Crabtree et al., 2012; Okagaki et al., 2011 and Okagaki et al., 2012

10. Lines 396-397 – The statement that Gpa1 phosphorylation is transient is unclear and not well supported.

11. Lines 415-421: It may be worth mentioning that the rim101 mutant, in which the GPA1/cAMP pathway is also upregulated but downstream of titan cell signaling therefore produces no titan cells, has an abnormal cell wall structure that also induces a Th1/Th17 immune response. Thus, it is possible that the fbp1 and CRK1OE strains are triggering the same cell wall phenotype but in the context of titan cells, resulting in an immune response that is similar to the rim101 mutant. See O'Meara et al., 2013; Ost et al., 2017; Wiesner et al., 2015

12. The manuscript had numerous issues with English language that made reading challenging. Instead of providing a comprehensive list of recommended changes, an edited version of the manuscript with some of the suggested modifications is attached. This is not an exhaustive editing of the manuscript, and it is recommended that the authors have a native English speaker review the manuscript for proper usage of scientific language prior to resubmission.

In their review of the first version of this manuscript, reviewer #1 added some comments to the manuscript file. These comments were forwarded to the authors, who replied as included in this Peer Review File.

Reviewer #2 (Remarks to the Author):

This manuscript investigates the control of cell size, in particular the formation of Titan cells, and its relation to fungal virulence in *C. neoformans*. It was previously shown that Gpa1-cAMP signaling regulates cell size and fungal virulence. Here the authors report that Gpa1 is itself regulated through phosphorylation by the protein kinase Crk1, and that Crk1 is regulated by the ubiquitin ligase SCF-Fbp1. Thus all three proteins (Gpa1, Fbp1 and Crk1) regulate meiosis, Titan cell formation, and fungal virulence.

The present manuscript would be much stronger if it showed that purified SCF-Fbp1 directly ubiquitinates Crk1 or at a minimum that Crk1 is actually ubiquitinated (and that Fbp1 is required for the appearance of ubiquitinated Crk1 in cells). It would also be

much stronger if the paper showed that purified Crk1 directly phosphorylates Gpa1 or at a minimum that Gpa1 is actually phosphorylated.

Key findings

1. Fbp1-Crk1 and Crk1-Gpa1 interactions are shown by yeast two hybrid and co-IP.
2. Deletion of FBP1 stabilizes Crk1, increases cell body size, and decreases virulence.
3. Deletion of CRK1 increases Gpa1 mobility (decreases phosphorylation?), decreases cell body size and increases virulence. Overexpression or stabilization of CRK1 has opposing effects.

Titan cell formation and virulence is shown in lungs during *Cryptococcus* infection, and is well quantified, for fbp1 mutant, crk1 mutant and CRK1 overexpression. Meiosis is shown by visualization of mating structures. This could be done in a more quantitative manner (FACS?).

Crk1 stability was shown by western blotting after a block on transcription, in WT and fbp1 mutant cells. Note: this experiment does NOT show that "Crk1 is a substrate of Fbp1," as claimed. This crucial experiment should be done in a more quantitative, direct, and well controlled, manner. Is Crk1 actually ubiquitinated? Does ubiquitination require Fbp1? Is this band really Crk1 (there is no negative control, such as untagged Crk1)?

Gpa1 phosphorylation was shown by a mobility shift upon western blotting in WT and crk1 mutant cells. NOTE: a mobility shift could be due to phosphorylation (or another modification) and as a result of Crk1 activity (or the activity of another downstream protein kinase or phosphatase). This crucial experiment should be done in a more quantitative, direct, and well controlled, manner. Is this band really Gpa1 (again there is no negative control for the antibody)? Is the mobility shift reversed by phosphatase treatment? By mutation of the phosphorylation site? By mutationally inactivating Crk1 kinase activity? Is it also affected by deletion of Fbp1? Why is much of Gpa1 still phosphorylated in the complete absence of the kinase that phosphorylates Gpa1?

More broadly speaking, I have a major problem with the mechanistic conclusions that the authors draw from non-mechanistic genetic approaches. The model in Figure 6 is not adequately supported by the evidence, or the central findings as stated in the title. As written, all we know is that loss of three different proteins (a ubiquitin ligase, a kinase, and a G protein) results in related cellular phenotypes. However a shared phenotype does not equal an enzyme-substrate relationship. Given that the gene expression profiles overlap only partially (Fig. 3) it seems that functions overlap only partially. The proposed mechanistic relationship could be direct or indirect, and in common or in parallel pathways. For example, in *S. cerevisiae* Elm1 phosphorylates Gpa1 but it also ubiquitinates other proteins that regulate cell growth independently of Gpa1. Cdc4 ubiquitinates Gpa1 but it also ubiquitinates other proteins that regulate cell growth independently of Gpa1.

Reviewer #3 (Remarks to the Author):

Cao et al., investigate the reduced dissemination and pathogenesis of fb1pD, lacking an E3 ligase previously shown to physically interact with the kinase Crk1. They nicely demonstrate that Crk1 is an Fbp1 substrate, that Crk1 phosphorylates Gpa1, and propose a model in which Fbp1 influences cell size and ploidy changes consistent with titan cell formation. These findings are further linked to the reduced virulence of a constitutively active GPA1Q284L mutant. Overall, these findings build on published work and expand our understanding of how Gpa1 and the cAMP pathway influence cell size in *C. neoformans*.

Despite this, I have several concerns that must be addressed, primarily pertaining to data analysis and presentation.

Major comments

Statistics: For any figure where values are compared, it is not appropriate to perform student's t-test for more than two samples. If the same control is used, then the analysis must adjust for multiple comparisons. In addition, the authors should ensure the data is normally distributed if the t-test is used. In the majority of these datasets, the data appear to be non-parametric. Where "1000 cells" are counted, the authors should specify whether this represents a pooled dataset across multiple biological replicates, or is a single biological replicate representative of multiple datasets. For % titan cells, the authors should specify whether these are technical or biological replicates in the legend. If only technical replicates, biological replicates are required. This in particular applies to all figures where % titan is reported, but should be considered wherever more than one comparison is made. For example, the authors as a regular practice do not perform a statistical analysis of complemented or OE strains, and this should be corrected.

Figure details throughout are scant, particularly where FP reporters are presented. For example, the tagged proteins should be indicated within the figure panel, not only in the legend, and the presence of and FP tag should be mentioned in the main text as part of the description of the experiment.

I'm a bit confused by data presented in figure 3 and S2, regarding Crk1OE. First, the authors appear to have omitted details about the construction of the OE strain and the Crk1-pest strain, including quantification of the degree of over-expression. Are these the same strains that are shown in figure S3b and S3C? Construct details should be included in the main text to improve clarity. Is this over expression constitutive, or induced? Details of promotor should be included in the main text and figure legends to improve clarity.

Second, it appears figure S2B right panel represents sorted or gated cells, but this is not clear from the text. If gated, the gating strategy should be provided. If sorted, the sorting strategy should be explained. However, if the black line represents an uninduced/control condition, this raises the question: are Crk1OE cells base diploid, even when grown on YPD (and OE induced, if needed)? Diploid cells are larger and produce larger titan cells and larger overall titan-induced populations.

Figure 4 H/I: This analysis ignores that titan cells are majority vacuole, with cytoplasm compressed to one side of the cell. Localization of GFP-GPA to the cell membrane requires higher resolution imaging and a marker to differentiate vacuole, membrane, and cytoplasm. Otherwise this analysis adds nothing to the manuscript and in fact oversells otherwise strong the findings. Line 348/349 in the discussion should be modified to avoid over-interpretation in the face of weak data.

More generally, I wonder if the authors have considered which aspect of titan cell induction is driving the observed phenotypes. The *in vitro* method of Hommel et al., can be understood as a combination of pH, hypoxia, nutrient starvation, and temperature, as well as membrane stress during shaking. The cAMP pathway, for example, is a key regulator of CO₂ response, and also regulates cell membrane stress responses. Have the authors examined how Fbp1 and Crk1 integrate or mediate response to one or more of these combinatorial signals?

Minor comments

Figure 1:
See comments about statistical analysis.

Figure 2:
The length of the "CT" domain should be indicated. Does this include PEST or not?

Figure 3:

Growth conditions used for the RNAseq experiment should be provided in the figure legend.

The RNA seq data should be made public upon publication (currently accession number is not provided).

Figure 4

C: what statistical analysis was performed (see line 254-256)

F: Clarity would be improved by indicating IP vs total protein on the figure rather than in the legend.

G: A t-test can only be performed on two samples. Which two samples were compared here? What about the other samples? It is preferable to use the appropriate test for the number of possible comparisons and correct for multiple comparisons to a control.

Figure S5c-d.

A t-test for four samples isn't correct, even if the samples look like they're similar. Please perform and report an appropriate analysis.

Figure 5:

It was nice to see the authors perform the correct statistical analysis for multiple comparisons with non-parametric data. Similar rigor should be applied throughout the manuscript.

Discussion line 408-410: Usv101 is a second example of a similar hypertitanising isolate associated with reduced dissemination.

Line 410-412: this would be further supported by Okagaki 2012 showing reduced uptake of yeast in the presence of titan cells (doi: 10.1128/EC.00121-12)

For all flow cytometry data, show gating strategies (for exclusion of doublets, sub-populations, etc).

What is the genotype of the CRK1OE strain? it is unclear from the text and apparently not included in the strain list. The details of the construction of this strain should also be included in the methods.

Figure S5E: What is GFP-tagged ? How were cells prepared for imaging? YPD can cause auto-fluorescence in the 488 (green) channel. Was background fluorescence excluded/controlled for?

line 95/96: incomplete sentence

line 130, 187, 193, etc: median not medium (assuming the median and not the mean is reported. For non-parametric populations, median is appropriate.) If the median was reported, then the authors should not use the word "average", as this implies mean.

line 148: terminus not terminal

197: significantly fewer, not less (and correct similar issues throughout).

Throughout: "titan inducing condition " not "titan inducible condition" Also: watch out for "tian".

Line 241: To improve clarity, specify for the reader that GPA1Q284L is constitutively active.

line 151; 177-179; 183, 217, 231/232, 249, 293-297, 308, 309, 310/311, 311, 344, 358, 374, 376, 415: typos

Title: Ubiquitin proteolysis of a CDK-related kinase regulates cell size 1 and fungal virulence in *Cryptococcus neoformans* (NCOMMS-21-47756)

REVIEWER COMMENTS

Reviewer #1 (Remarks to the Author) [please see also the attached document]:

The manuscript by Cao and colleagues describes the function of the E3 ligase Fbp1 in titan cell formation and how its ubiquitination of the CDK-related kinase Crk1 results in control of Gpa1 and cAMP/PKA signaling that ultimately leads to cell size modifications and titan cell formation in the human fungal pathogen *Cryptococcus neoformans* that impact virulence. These studies provide improved clarity on critical unknown aspects of the signal transduction pathway involved in cell size regulation that link cell division and meiosis events in a context outside of classical mating/sporulation – a process that is not well understood in any organism but that is of critical importance, especially in the context of human cancer, adaptation, evolution, and even drug resistance mechanisms.

The studies presented are clear and well described with only minor experimental areas where the authors have overstated their conclusions or where the data are not thoroughly supported by their data. Detailed comments are listed below.

1. Example H&E sections that were analyzed to generate the data presented in Fig 1a need to be included so the reader can determine whether the data presented accurately represent the data that are visible in the microscopic sections. Because these microscopic slides are typically only 5µm tissue sections and *Cryptococcus* cells are anywhere from 2-100µm in diameter, using slides for measurements of *Cryptococcus* size has a high degree of inaccuracy because it is unknown if the measurement is being taken through the middle of the cell. If serial sections are not analyzed to verify that the same cell is not analyzed on multiple slides, and that all cells have an equal chance of being represented in sections, then an indication of cell size can be gleaned from analysis of H&E sections. However, visual comparison of the slides and the data should always be presented simultaneously.

A: Thank you for this suggestion. We have included examples of H&E sections in the figure s1a to represent the titan cell percentage of H99, the *fbp1Δ* mutant and the *fbp1Δ+FBP1* complement strain. Example of H&E-stained slides can also be found in our previous published paper (DOI: [10.1128/IAI.00994-13](https://doi.org/10.1128/IAI.00994-13)). As figure s1a shows, our tissue sections were over 5µm. We observed abundant H99 and the *fbp1Δ+FBP1* cells in infected lungs, but lungs infected by the *fbp1Δ* mutant showed very few yeast cells at different time points. We analyzed serial sections of lung tissues from different mice to avoid repeat measurement of the same cell. We agree that measurements of *Cryptococcus* size using slides has a high degree of inaccuracy, and not all cells were measured from the middle, we thought that all three infected lung samples were

measured by the same way, the data should be acceptable. We also confirmed this observation by quantification of large cell percentage in BALF samples (Fig. 1b-1d).

2. This reviewer has concerns with the way the statistics are presented in Figure 1. As presented, statistical significance was only observed between the wt and *fbp1* mutant and not between the mutant and the complement, even though the complement appears to have a similar phenotype to the wt. If this is true, these data are very concerning and raises the question as to how statistically significance was obtained for the mutant, but not the complement in comparison to the wt, and why a difference between the mutant and the complement was also not observed. These statistics do not make any sense to this reviewer.

A: Thanks for this important comment and sorry for the confusion in statistics presentation. In original submission, we only compared wild type with the *fbp1*Δ mutant using two-tailed t test because all our previous studies showed that the mutant can be fully complemented by reintroducing *FBP1* gene. In this revision, we corrected the statistical analysis with the Kruskal-Wallis nonparametric test for multiple comparison. Here we present the statistical significance between wild type and all other groups.

3. Fig 1f – indicate the difference between the black and red lines in the flow plot (assume the black line is normal conditions and the red line is in BAL but this is not stated in the figure legend or on the diagram).

A: We apologize for the incomplete information in our original submission. We added the figure s1d to show the gating strategy during the analysis of the DNA content. Briefly, cells were cultured under titan cell inducing conditions for 3 days and analyzed by dot plots (FSC/SSC) using flow cytometry. FSC/SSC^{high} (red borders/lines) and FSC/SSC^{low} (black borders/lines) represent titan cells (TC) and typical cells (tC), respectively. This data is to confirm that the large cells from the *fbp1*Δ mutant has the same polyploidy as those in the wild type H99.

4. Lines 177-180. This section needs to be rewritten as the authors step over the line between summarizing their results and making conclusions. While they can state that their data suggest that *Crk1* plays important role in *Fbp1*-mediated fungal meiosis and sporulation. They provide no direct data in this section to link *Crk1* to vegetative growth or titan cell formation. Thus, their last sentence is speculation. It would be better to lay the foundation for this argument in the previous paragraph and then more fully develop this concept in the Discussion section.

A: Thank you for pointing out the misleading sentence. This is a valid point. The last sentence is not a conclusion, it is our observation from Fig 2d. We have deleted this sentence and mentioned this information in the discussion part “We also identified a *Crk1* signal when cells were grown in YPD culture conditions (Fig 2d), indicating that

Crk1 is not a meiosis-specific protein and may exhibit cellular functions other than regulating meiosis processes in *C. neoformans*, which is different from its Ime2 homolog in *S. cerevisiae*^{56,59}.

5. Fig s2b should be moved to Fig3 as this information is critical to show that the large cells generated by the CRK1OE strain are bona fide titan cells.

A: Thank you for this suggestion. We have moved FACS analysis of DNA content in the CRK1^{OE} and the CRK1^{ΔPEST} strains to the main figure fig.1e.

6. Fig 3h. “Categories” is vague. Are these GO terms? Metabolic categories?

A: Sorry that we did not make the categories clear. It is based on GO term. We added the information to the figure legend of Fig.3i and the main text (line 226).

7. The *crk1Δ GPA1Q284L* mutant grown under in vitro titan inducing conditions that produces enlarged cells needs to be analyzed for ploidy to confirm that the cells are bona fide titan cells and the data presented in the figure. Similarly, all of the strains in Fig 4 that are putatively producing titan cells should be analyzed for ploidy to confirm titan cell formation in this analysis and the data presented in the figure.

A: Thank you for this suggestion. We analyzed the ploidy of enlarged cells that are induced under the titan cell inducing conditions. As shown in Fig s6d, the *GPA1Q284L* strain, the *crk1Δ GPA1Q284L* strain, and the strains treated with exogenous cAMP or overexpression of *GPA1* can be induced to produce polyploidy in enlarged cells.

8. The localization data for Gpa1 is unconvincing. It all hinges on fluorescence data presenting in Figure 4i. Yet it is unclear how the authors accounted for the difference in volume/size of the cells in their determination of the fluorescence and localization of the GPA1:GFP signal. Could the decrease in cytosolic signal in the titan cell simply be due to the larger size of the titan cell resulting in the protein being distributed across a larger volume (i.e. a bigger sphere). What happens if the authors take into account the volume of the cell when measuring the internal/cytosolic fluorescence? In the Discussion section, Lines 348-350, the authors make the bold statement that “Crk1-mediated Gpa1 phosphorylation induces Gpa1 membrane localization to trigger GPCR activation of the Gpa1-cAMP signaling to regulate cell size in *C. neoformans*”. In this reviewer’s opinion, the authors cannot make this statement without significant additional proof of differences in Gpa1 protein localization differences in the Crk1 mutants. For example, the authors could perform sub-cellular localizations with Gpa1-FLAG strain in the different backgrounds under the titan inducing conditions and examine the membrane vs. cytosolic fractions to quantify protein levels in the different fractions.

A: This is a valid point. Besides the localization of the GPA1:GFP signal, we measured

the sub-cellular localization of Gpa1 signal in the typical size cells and titan cells in a western blot as suggested. Our data confirmed that more Gpa1 membrane localization in the titan cell enriched samples. Based on the result (Fig. 4I), we have added the following sentences to the main text. “Since the *CRK1 Δ ^{PEST}* strain produced more titan cells than did the *crk1 Δ* mutant when overexpressing *GPA1*, cells of the *crk1 Δ* mutant and the *CRK1 Δ ^{PEST}* strain expressing P_{GDP1}-Gpa1:FLAG were used to represent typical cells and titan cells, respectively. We separated the membrane and cytosolic fractions of typical cells and titan cells. Samples were normalized to the same dry weight and were analyzed for the presence of the Gpa1:FLAG signal in each fraction. As shown in Fig 4I, typical cells and titan cells exhibit comparable Gpa1 levels in the cytosolic fraction, whereas titan cells exhibit more Gpa1 in the membrane fraction compared to typical cells.”

9. Line 321 – overproduction of titan cell has previously been shown to result in decreased virulence. See Crabtree et al., 2012; Okagaki et al., 2011 and Okagaki et al., 2012

A: We cited these three publications in our discussion part. “Virulence attenuation of *CRK1 Δ ^{PEST}*, *fbp1 Δ* , and *GPA1^{Q284L}* strains is paradoxical to their overproduction of titan cells because titan cells are known to contribute to cryptococcal virulence (Okagaki et al, 2010; Zaragoza et al, 2010; Okagaki et al, 2012; Okagak et al, 2011 Crabtree et al, 2012)” (line 440-442)

10. Lines 396-397 – The statement that Gpa1 phosphorylation is transient is unclear and not well supported.

A: Thank you for pointing out the misleading sentence. We have corrected this sentence to “It should be noted that Gpa1 phosphorylation by Crk1 was detected under titan cell inducing conditions, but not in the other conditions we tested, thereby indicating a specific role of Crk1 in the regulation of titan cell production.” (line 429-432)

11. Lines 415-421: It may be worth mentioning that the *rim101* mutant, in which the GPA1/cAMP pathway is also upregulated but downstream of titan cell signaling therefore produces no titan cells, has an abnormal cell wall structure that also induces a Th1/Th17 immune response. Thus, it is possible that the *fbp1* and *CRK1OE* strains are triggering the same cell wall phenotype but in the context of titan cells, resulting in an immune response that is similar to the *rim101* mutant. See O’Meara et al., 2013; Ost et al., 2017; Wiesner et al., 2015

A: Thank you for the suggestion. We have added several sentences in our discussion. “Indeed, the studies on Rim101, a downstream transcription factor of cAMP/PKA pathway, show that deletion of *RIM101* alters cell surface structure and blocks titan cell production {Ost, 2017 #2179} {O’Meara, 2013 #2181}. The cell surface component chitin has been reported to both stimulates and inhibits immune responses {Ost, 2017

#2179}{Wiesner, 2015 #1608}. The *rim101*Δ mutant helps to expose cell surface antigens to the immune system and increase proinflammatory cytokine levels, including strong Th1 and Th17 responses {Ost, 2017 #2179}. The *fbp1*Δ mutant and the *CRK1*^{ΔPEST} strain may have similar cell surface changes as the *rim101*Δ mutant, and these possibilities need to be investigated in future studies.” (line454-461)

12. The manuscript had numerous issues with English language that made reading challenging. Instead of providing a comprehensive list of recommended changes, an edited version of the manuscript with some of the suggested modifications is attached. This is not an exhaustive editing of the manuscript, and it is recommended that the authors have a native English speaker review the manuscript for proper usage of scientific language prior to resubmission.

A: We are grateful for the careful edits and the recommended modifications by the reviewer. The revised version has been carefully read by all authors and a native English speaker to correct any grammatical errors.

Reviewer #2 (Remarks to the Author):

This manuscript investigates the control of cell size, in particular the formation of Titan cells, and its relation to fungal virulence in *C. neoformans*. It was previously shown that Gpa1-cAMP signaling regulates cell size and fungal virulence. Here the authors report that Gpa1 is itself regulated through phosphorylation by the protein kinase Crk1, and that Crk1 is regulated by the ubiquitin ligase SCF-Fbp1. Thus all three proteins (Gpa1, Fbp1 and Crk1) regulate meiosis, Titan cell formation, and fungal virulence.

The present manuscript would be much stronger if it showed that purified SCF-Fbp1 directly ubiquitinates Crk1 or at a minimum that Crk1 is actually ubiquitinated (and that Fbp1 is required for the appearance of ubiquitinated Crk1 in cells). It would also be much stronger if the paper showed that purified Crk1 directly phosphorylates Gpa1 or at a minimum that Gpa1 is actually phosphorylated.

A: We thank the reviewer for these insightful suggestions. We have conducted the *in vivo* ubiquitination assay (Fig. 2g), phosphatase assay (Fig. 4f) and *in vitro* kinase assay (Fig.4g) to support our conclusions. We also repeated the detection of Gpa1 phosphorylation during titan cell induction with additional controls as suggested. Our results confirmed that Crk1 is ubiquitinated by Fbp1 E3 ligase, and Crk1 directly phosphorylates Gpa1. The manuscript has been updated with detailed presentation of the new data.

Key findings

1. Fbp1-Crk1 and Crk1-Gpa1 interactions are shown by yeast two hybrid and co-IP.
2. Deletion of FBP1 stabilizes Crk1, increases cell body size, and decreases virulence.

3. Deletion of CRK1 increases Gpa1 mobility (decreases phosphorylation?), decreases cell body size and increases virulence. Overexpression or stabilization of CRK1 has opposing effects.

Titan cell formation and virulence is shown in lungs during *Cryptococcus* infection, and is well quantified, for *fbp1* mutant, *crk1* mutant and CRK1 overexpression. Meiosis is shown by visualization of mating structures. This could be done in a more quantitative manner (FACS?).

A: We agree with the reviewer that it is better to show meiosis in a more quantitative manner. However, we will not be able to quantify meiosis efficiency in *C. neoformans* due to the technique limitations. *C. neoformans* belongs in the phylum Basidiomycota, because it generates a filamentous sexual state that results in spore production from a basidium structure. Mating process in *C. neoformans* includes cells fusion, dikaryotic filament formation, basidium formation, nuclear fusion, and then meiosis. Unlike many ascomycetes that undergo meiosis to produce four spores, *Cryptococcus* species are distinguished by basidia with four long chains of attached spores that are easily dispersed. Therefore, it is difficult to isolate the meiosis specific material for quantification. In addition, mating only happens on solid agar plates or other hard surface, adding to the challenge to isolate meiosis specific materials. We often perform cell fusion assay to quantify the cell fusion efficiency in yeast form, but that step is prior to meiosis. We would certainly be very interested in finding a way to quantify meiosis in *C. neoformans*, but may be beyond the scope of this study.

Crk1 stability was shown by western blotting after a block on transcription, in WT and *fbp1* mutant cells. Note: this experiment does NOT show that “Crk1 is a substrate of Fbp1,” as claimed. This crucial experiment should be done in a more quantitative, direct, and well controlled, manner. Is Crk1 actually ubiquitinated? Does ubiquitination require Fbp1? Is this band really Crk1 (there is no negative control, such as untagged Crk1)?

A: This is a valid point. We thank the reviewer for this suggestion. We did the *In vivo* ubiquitination assay to detect the polyubiquitinated Crk1. As shown in Figure 2g. The smear signal on the top of Crk1:HA (~141kD) indicates that Crk1 is ubiquitinated. We treated cells with proteasome inhibitor MG132, purified Crk1 using HA antibody and detected Crk1:HA signal using HA antibody. We observed that Crk1 ubiquitination is diminished in the *fbp1*Δ mutant compared to the H99 background, which suggests that Fbp1 is involved in Crk1 ubiquitination. We did not detect HA signal in negative control (H99+vector), which confirms that the band is real Crk1:HA signal.

Gpa1 phosphorylation was shown by a mobility shift upon western blotting in WT and *crk1* mutant cells. NOTE: a mobility shift could be due to phosphorylation (or another modification) and as a result of Crk1 activity (or the activity of another downstream protein kinase or phosphatase). This crucial experiment should be done in a more

quantitative, direct, and well controlled, manner. Is this band really Gpa1 (again there is no negative control for the antibody)? Is the mobility shift reversed by phosphatase treatment? By mutation of the phosphorylation site? By mutationally inactivating Crk1 kinase activity? Is it also affected by deletion of Fbp1? Why is much of Gpa1 still phosphorylated in the complete absence of the kinase that phosphorylates Gpa1?

A: Thank you for these suggestions. We performed the phosphatase assay (Figure 4f) and observed that the mobility shift of either total extracts or purified Gpa1:FLAG can be reversed by phosphatase treatment. In Figure 4h, we added a negative control (H99) and didn't detect any FLAG signal using anti-FLAG antibody. We also detected Gpa1:FLAG signal in the *fbp1*Δ mutant and the *CRK1*^{ΔPEST} background strains. The quantitative data showed no significant difference compared to the H99 background.

In our revised manuscript, we didn't mutant the phosphorylation site of Gpa1. Although CnGpa1 is the homolog of Gpa2 in *S. cerevisiae*, the phosphorylation sites of CnGpa1 remain unclear. The direct phosphorylation sites of Crk1 in CnGpa1 will be a good direction for future investigation. We also performed the *in vitro* kinase assay to show that Crk1 phosphorylates Gpa1 in the presence of ATP (Figure 4g).

Low levels of phosphorylated Gpa1 and titan cell production could still be detected in the absence of Crk1, suggesting that other protein kinases are likely also involved in Gpa1 phosphorylation in *C. neoformans*. For example, it has been published that glycogen synthase kinase (GSK) phosphorylates Gpa2 under nitrogen starvation conditions in *S. cerevisiae*. (doi: 10.1074/jbc.RA119.009609)

More broadly speaking, I have a major problem with the mechanistic conclusions that the authors draw from non-mechanistic genetic approaches. The model in Figure 6 is not adequately supported by the evidence, or the central findings as stated in the title. As written, all we know is that loss of three different proteins (a ubiquitin ligase, a kinase, and a G protein) results in related cellular phenotypes. However a shared phenotype does not equal an enzyme-substrate relationship. Given that the gene expression profiles overlap only partially (Fig. 3) it seems that functions overlap only partially. The proposed mechanistic relationship could be direct or indirect, and in common or in parallel pathways. For example, in *S. cerevisiae* Elm1 phosphorylates Gpa1 but it also ubiquitinates other proteins that regulate cell growth independently of Gpa1. Cdc4 ubiquitinates Gpa1 but it also ubiquitinates other proteins that regulate cell growth independently of Gpa1.

A: This is a valid point. We agree with the reviewer that the proposed mechanistic relationship could be direct or indirect. Our study showed that Fbp1 functions as an E3 ligase, while Crk1 functions as a CDK-related kinase. Both proteins are involved in numerous protein degradation or modification, and the Gpa1-cAMP pathway regulates multiple biological processes. Based on the suggestions, we performed some mechanistic genetic assays, including the *in vivo* ubiquitination assay, phosphatase

assay and *In vitro* kinase assay, to provide more direct biochemical evidence to support our conclusion. Consistent with the model, our data demonstrate that Crk1 ubiquitination is dependent on Fbp1, and Gpa1 can be phosphorylated by Crk1. Crk1 connects the role of ubiquitin-proteasome system and the cAMP pathway to regulate titan cell production help to understand the mechanism of cell size and pathogenesis in *C. neoformans*.

Reviewer #3 (Remarks to the Author):

Cao et al., investigate the reduced dissemination and pathogenesis of *fbp1D*, lacking an E3 ligase previously shown to physically interact with the kinase Crk1. They nicely demonstrate that Crk1 is an Fbp1 substrate, that Crk1 phosphorylates Gpa1, and propose a model in which Fbp1 influences cells size and ploidy changes consistent with titan cell formation. These findings are further linked to the reduced virulence of a constitutively active GPA1Q284L mutant. Overall, these findings build on published work and expand our understanding of how Gpa1 and the cAMP pathway influence cell size in *C. neoformans*.

Despite this, I have several concerns that must be addressed, primarily pertaining to data analysis and presentation.

Major comments

Statistics: For any figure where values are compared, it is not appropriate to perform student's t-test for more than two samples. If the same control is used, then the analysis must adjust for multiple comparisons. In addition, the authors should ensure the data is normally distributed if the t-test is used. In the majority of these datasets, the data appear to be non-parametric. Where "1000 cells" are counted, the authors should specify whether this represents a pooled dataset across multiple biological replicates, or is a single biological replicate representative of multiple datasets. For % titan cells, the authors should specify whether these are technical or biological replicates in the legend. If only technical replicates, biological replicates are required. This in particular applies to all figures where % titan is reported, but should be considered wherever more than one comparison is made. For example, the authors as a regular practice do not perform a statistical analysis of complemented or OE strains, and this should be corrected.

A: Thank you for pointing out this error. We have corrected the statistical analysis. "Statistical analysis was performed with the Kruskal-Wallis nonparametric test for multiple comparisons and Mann-whitney nonparametric test for two samples." For cell size measurement or % titan cells, we specified the dataset and biological replicates. Please find the details in the figure legends.

Figure details throughout are scant, particularly where FP reporters are presented. For example, the tagged proteins should be indicated within the figure panel, not only in the legend, and the presence of and FP tag should be mentioned in the main text as part of the description of the experiment.

A: Thank you for pointing out this missing information. We added the Gpa1:GFP in Figure 4i and mention the GFP tag in the main text. “We investigated the subcellular localization of GFP-fused Gpa1 expressed from the *ACT1* promoter in H99, *crk1Δ*, and *crk1Δ+CRK1* strain backgrounds.”

I'm a bit confused by data presented in figure 3 and S2, regarding Crk1OE. First, the authors appear to have omitted details about the construction of the OE strain and the Crk1-pest strain, including quantification of the degree of over-expression. Are these the same strains that are shown in figure S3b and S3C? Construct details should be included in the main text to improve clarity. Is this over expression constitutive, or induced? Details of promoter should be included in the main text and figure legends to improve clarity.

A: Thank you for pointing out this incomplete information. We have added strain details when they were first mentioned in the main text (CRK1OE: line 178 and line 188. CRK1ΔPEST: line196) and also the details of strain generation in the material and methods section (line 644-648).

Second, it appears figure S2B right panel represents sorted or gated cells, but this is not clear from the text. If gated, the gating strategy should be provided. If sorted, the sorting strategy should be explained. However, if the black line represents an uninduced/control condition, this raises the question: are Crk1OE cells base diploid, even when grown on YPD (and OE induced, if needed)? Diploid cells are larger and produce larger titan cells and larger overall titan-induced populations.

A: We apologize for not making it clear in our original submission. Lines represent gating cells under titan cell inducing conditions. We added the Figure s1d to show the gating strategy during the analysis of the DNA content. Briefly, Cells were cultured under titan cell inducing conditions for 3 days and analyzed by dot plots (FSC/SSC) using flow cytometry. FSC/SSC^{high} (red borders/lines) and FSC/SSC^{low} (black borders/lines) represent titan cells (TC) and typical cells (tC), respectively.

Figure 4 H/I: This analysis ignores that titan cells are majority vacuole, with cytoplasm compressed to one side of the cell. Localization of GFP-GPA to the cell membrane requires higher resolution imaging and a marker to differentiate vacuole, membrane, and cytoplasm. Otherwise this analysis adds nothing to the manuscript and in fact oversells otherwise strong the findings. Line 348/349 in the discussion should be modified to avoid over-interpretation in the face of weak data.

A: This is a valid point. Reviewer #1 also expressed the similar concern. To address this concern and confirm the shift of Gpa1 localization, we performed the sub-cellular localizations of Gpa1 in the typical size cells and titan cells. We separated the membrane and cytosolic fractions of typical cells and titan cells detected the Gpa1:FLAG signal of each fraction. As shown in Fig 4l, typical cells and titan cells have comparable amount of Gpa1 in the cytosolic fractions, while titan cells have more Gpa1 than typical cells in the membrane fractions. This results are consistent with our interpretation that Crk1 phosphorylation of Gpa1 induces its membrane localization during titan cell induction.

More generally, I wonder if the authors have considered which aspect of titan cell induction is driving the observed phenotypes. The *in vitro* method of Hommel et al., can be understood as a combination of pH, hypoxia, nutrient starvation, and temperature, as well as membrane stress during shaking. The cAMP pathway, for example, is a key regulator of CO₂ response, and also regulates cell membrane stress responses. Have the authors examined how Fbp1 and Crk1 integrate or mediate response to one or more of these combinatorial signals?

A: Thank you for this suggestion. We have not examined how Fbp1 and Crk1 integrate or mediate response to one or more of these combinatorial signals. Previous studies identified two G protein coupled receptors (Ste3 and Gpr5) are involved in titan cell induction, but the ligands for these receptors remain unclear. In this manuscript, we are focusing on the role of Fbp1 in fungal virulence and try to find its substrates that are involved in Fbp1 mediated fungal pathogenesis. To make the quantification easier and more accurate, we tried the *in vitro* titan cell inducing conditions generated by Hommel et al. Identifying signals regulating titan cell formation is a highly interesting topic that remains to be investigated.

Minor comments

Figure 1:

See comments about statistical analysis.

A: Corrected. Thanks

Figure 2:

The length of the "CT" domain should be indicated. Does this include PEST or not?

A: Thank you for pointing out. We indicated the length of the "CT" in figure 2a. It includes the PEST domain.

Figure 3:

Growth conditions used for the RNAseq experiment should be provided in the figure legend.

A: Growth conditions have been added into the legend of fig 3g

The RNA seq data should be made public upon publication (currently accession number is not provided).

A: The accession number (PRJNA816899) was provided.

Figure 4

C: what statistical analysis was performed (see line 254-256)

A: “Statistical analysis was performed with the Kruskal-Wallis nonparametric test for multiple comparisons. **, $P \leq 0.01$.” was included in the figure legend.

F: Clarity would be improved by indicating IP vs total protein on the figure rather than in the legend.

A: Thank you for this suggestion, we indicated IP and total proteins on the figure 4e.

G: A t-test can only be performed on two samples. Which two samples were compared here? What about the other samples? It is preferable to use the appropriate test for the number of possible comparisons and correct for multiple comparisons to a control.

A: We corrected the statistical analysis with the Kruskal-Wallis nonparametric test

Figure S5c-d.

A t-test for four samples isn't correct, even if the samples look like they're similar. Please perform and report an appropriate analysis.

A: We corrected the statistical analysis for multiple comparisons

Figure 5:

It was nice to see the authors perform the correct statistical analysis for multiple comparisons with non-parametric data. Similar rigor should be applied throughout the manuscript.

A: Thank you. We corrected all the statistical analysis in the figures.

Discussion line 408-410: Usv101 is a second example of a similar hypertitanising isolate associated with reduced dissemination.

Line 410-412: this would be further supported by Okagaki 2012 showing reduced uptake of yeast in the presence of titan cells (doi: 10.1128/EC.00121-12)

A: Thank you for these suggestions, we mentioned these papers and cited them.

For all flow cytometry data, show gating strategies (for exclusion of doublets, sub-populations, etc).

A: We showed the gating strategies for the flow cytometry data. Please see the figures s1d and s3b-e.

What is the genotype of the CRK1OE strain? it is unclear from the text and apparently not included in the strain list. The details of the construction of this strain should also be included in the methods.

A: Sorry that we didn't make the strain information clear. We used two *CRK1^{OE}* strains in our study. CUX151 (MATalpha *P_{HIS}-GFP:CRK1:HA-NAT*) was used for mating and CUX1291 (MATalpha *P_{ACT1}-CRK1:mCherry-NAT*) was used for other experiments. We added the genotype of strains when they were first time mentioned in the text. The details of strain generation were included in the method "Fluorescence imaging"

Figure S5E: What is GFP-tagged ? How were cells prepared for imaging? YPD can cause auto-fluorescence in the 488 (green) channel. Was background fluorescence excluded/controlled for?

A: The images show the GFP tagged Gpa1. We have added the Gpa1:GFP in figure s6g. The details of cells preparation for imaging were included in the material and methods "Gpa1:GFP localization." We noticed auto-fluorescence in the green channel during imaging and added one image in figure s6g to show the background fluorescence. We also and confirmed our GFP-tagged strains by western blotting using the GFP antibody.

line 95/96: incomplete sentence

A: Corrected. Thanks

line 130, 187, 193, etc: median not medium (assuming the median and not the mean is reported. For non-parametric populations, median is appropriate.) If the median was reported, then the authors should not use the word "average", as this implies mean.

A: Corrected. Thanks

line 148: terminus not terminal

A: Corrected. Thanks

197: significantly fewer, not less (and correct similar issues throughout).

Throughout: "titan inducing condition " not "titan inducible condition" Also: watch out for "tian".

A: Corrected. Thanks

Line 241: To improve clarity, specify for the reader that GPA1Q284L is constitutively active.

A: Thank you for pointing out. We specified *GPA1^{Q284L}* as the strain expressing the *GPA1* dominant active allele (line 248).

line 151; 177-179; 183, 217, 231/232, 249, 293-297, 308, 309, 310/311, 311, 344, 358, 374, 376, 415: typos

A: All corrected. Thanks

Finally, we thank all reviewers for their constructive and insightful comments that help significantly improved our manuscript.

Reviewer #1 (Remarks to the Author):

The authors provided appropriate revisions that alleviate my previous concerns.

Reviewer #2 (Remarks to the Author):

The authors have done an adequate job of toning down their mechanistic conclusions and have added several mechanistic experiments in response to my previous comments.

First they tested whether Crk1 ubiquitination is dependent on Fbp1 and observed that the ubiquitination level of Crk1 is diminished in the fbp1 Δ mutant compared to the H99 background (Fig. 2g).

However they don't really know that the shifted band is ubiquitinated protein, it could be another modification entirely. I think the phrase "putatively" or "likely" ubiquitinated would be appropriate here. I would like to see documentation of reproducibility and quantitation of abundances of the modified and unmodified Crk1.

Second, to test whether Crk1 is capable of directly phosphorylating Gpa1, they purified Crk1:HA and Gpa1:FLAG and performed an in vitro kinase assay. They observed the slower migrating form of Gpa1 in the presence of both ATP and Crk1:HA (Fig. 4g), indicating that Crk1 directly phosphorylates Gpa1.

Again there is a heavy reliance on shifted bands, in one case it is inferred to be ubiquitin and the other it is inferred to be phosphorylation.

***** I do not agree that this experiment is detecting phosphorylation of Gpa1-Flag since a prominent Gpa1-Flag band is observed in the lane lacking Gpa1-Flag. *****

This phosphorylation experiment is lacking other essential controls. I would like to see Gpa1 plus ATP without Crk1, or better yet with catalytically inactive Crk1 mutant, to prove that it is purified and activated Crk1 that is the kinase and not a contaminant kinase.

Also since purified protein is being used the authors should show evidence of purity, by Coomassie staining of purified Crk1 and Gpa1.

Again, I would like to see documentation of reproducibility and quantitation of abundances of the modified and unmodified protein.

Reviewer #3 (Remarks to the Author):

I thank the authors for their effort to address the comments raised by this reviewer. Overall, I found the revised manuscript substantially improved. I applaud the authors for their clear presentation of the data and for sharing data in ways that are usable by others (Table S1). Overall, the work contributes significantly to an improved understanding of the molecular mechanisms governing titan cell formation and pathogenesis. Despite this there are still some issues that need to be addressed before the manuscript is suitable for publication.

Panel 3C CRK2 Δ PEST is not discussed. I suggest adding it at line 205.

Supplemental figures are well presented and contribute to understanding of the work. Supplemental Tables need clear, descriptive legends.

It is difficult to see how statements in line 267-269 suggesting that exogenous cAMP increases titan cell production in all strains are supported by the data presented in figure S6b, were there is no significant difference in median cell size for any strains except crk1. It would be better to present these data as scattered points, similar to S6c.

Is Figure S6D (top and bottom panels) not discussed in the text? This figure is very confusing. The referenced data is apparently the two panels floating off to the side on the right? This layout needs to be improved for clarity. Also, it's unclear what the blue lines in plot WT+cAMP indicate, as these do not match 1C/2C peaks and lack labels.

Data shown in Figure 4i,j S6g significantly improve the manuscript.

Figure 4K: it would be better to include the GPA1OE genotype on this figure panel.

In figure 4I, it is disingenuous to change mutant names to "typical" or "titan", especially since these terms are used for subpopulations throughout the rest of the manuscript. Were these cells fractionated by size? If not, then simply report strain names and explain the logic to your reader. Please make this correction in the text at lines 332, 334 as well.

Symbols on Figure 5a don't reproduce well in black and white. It is hard to differentiate WT and CRK1dPEST.

Figure 5B is not quantitative, but is described in the text to show that particular strains showed an "increased presence". This must be corrected by presenting quantitative data, as has been shown in other points in the manuscript.

Line 365/366, 452-454, 455-458, Multiple other authors have already shown that increased titan cell production correlates with diminished fungal virulence, as has already been pointed out to the authors. Statements that over emphasize the novelty of the authors findings undermine the overall work and should be revised. Others who have done work supporting this conclusion, in addition to work on Pdr802, should be cited, as suggested previously.

There are still some typos that need to be corrected, and some which have been incorrectly introduced in the new version, perhaps on the advice of others. I can confirm that I have a degree in English writing and grammar (in addition to being a "native" speaker) and that these suggestions are grammatically correct:

Line 56 "elongated hyphae cells" change to: elongated hyphal cells

Line 63 "cellular heterogeneity ... are often observed" change to: cellular heterogeneity ... is often observed

Line 81 "support fine-tune cell..." change to: support fine-tuned cell size regulation

Line 92 "mutant persists low levels.." change to: mutant persists at low levels...

Line 95 "low pulmonary persistency...." change to: low pulmonary persistence

Line 101 "The Gpa1-cAMP pathway activation": "The" is not needed, though this is mostly stylistic.

Line 105/106: "the Gpa1 G protein signal pathway through a kinase regulator Crk1" change to: the Gpa1 G protein signalling pathway through a kinase regulator, Crk1.

Line 119: lower case w in "we"

Line 148: I think "support" is a more accurate word to use than "perform"

Line 288-289 "is accounted" change to: To determine whether Crk1 activity accounts for Gpa1 phosphorylation....

Line 296: "proteins from the H99..." change to: proteins from H99...

Line 305: this needs to be two separate sentences and the second sentence needs correction:

We isolated the total protein from H99 expressing Gpa1:FLAG that were cultured in YPD overnight. The sample was treated with protein phosphatase to remove phosphorylation from Gpa1 before purification.

I suggest rewriting more simply:

Total protein was collected from overnight YPD cultures of H99 expressing Gpa1:FLAG and treated with protein phosphatase to remove phosphorylation before purification. We observed the slower migrating form of Gpa1 only in fractions treated subsequently with both ATP and Crk1:HA (Fig. 4g), indicating that Crk1 directly phosphorylates Gpa1.

Line 309-311: "Total proteins... were isolated" change to: Total protein ... was isolated (yes, English is weird)

Line 315: "or a rich medium of YPD" change to: "rich medium (YPD)" or "YPD rich medium"

Line 337: comma, not semi colon.

Line 339: "levels... and ...production is still" change to: levels... and ...production are still

Line 342: "in vivo and mice" change to: "in vivo, and mice

Line 350: "infection with Crk1OE strain" change to: infection with the Crk1OE strain

Line 351: "in infected lungs sections" change to: in infected lung sections

Line 352: "the lung of mice..." change to: the lungs of mice

Line 429: "functions upstream of the Gpa1-" change to: functions upstream of Gpa1-

Line 469: "has been reported to both stimulates and inhibits" change to "stimulate and inhibit"

Line 1109: "proportion in cells overexpression of GPA1" change to: proportion of cells overexpressing GPA1"

Figure legends also need to be read carefully for grammatical issues.

Title: Ubiquitin proteolysis of a CDK-related kinase regulates cell size and fungal virulence in *Cryptococcus neoformans* (NCOMMS-21-47756)

REVIEWER COMMENTS

Reviewer #1 (Remarks to the Author):

The authors provided appropriate revisions that alleviate my previous concerns.

Reviewer #2 (Remarks to the Author):

The authors have done an adequate job of toning down their mechanistic conclusions and have added several mechanistic experiments in response to my previous comments.

A: We thank the reviewer for the positive response to this revision.

First they tested whether Crk1 ubiquitination is dependent on Fbp1 and observed that the ubiquitination level of Crk1 is diminished in the *fbp1*Δ mutant compared to the H99 background (Fig. 2g).

However they don't really know that the shifted band is ubiquitinated protein, it could be another modification entirely. I think the phrase "putatively" or "likely" ubiquitinated would be appropriate here. I would like to see documentation of reproducibility and quantitation of abundances of the modified and unmodified Crk1.

A: Although we are confident that the shifted bands that lead to smear signal are ubiquitinated proteins, we agree with the reviewer that we cannot exclude the possibility of other post-translational modifications (PTM), such as myristoylation, phosphorylation, etc. But we expect most of the other PTMs would lead to single band shift, so would not fully explain the smear

signal we detected. We used “putatively ubiquitinated Crk1” in this version (line 178) as suggested by the reviewer.

We have done this assay multiple times and the results are reproducible. We have listed the following past results as examples (Image 1-3).

Image 1. Crk1 ubiquitination. To confirm the ubiquitination of Crk1, we detected ubiquitinated proteins in a western blot using **ubiquitin antibody** (Invitrogen, 14-6078-82) after immunoprecipitation with HA antibody (Genscript, A01244). Secondary antibody: Anti-mouse.

Image 2. Crk1 ubiquitination. Putatively ubiquitinated Crk1 was detected as a ladder-like smear with high-molecular-weight protein in a western blot using **HA antibody** (Genscript, A01244). Secondary antibody: Anti-mouse.

Image 3. Crk1 ubiquitination (over-exposed western results show smear signal). Putatively ubiquitinated Crk1 was detected as a ladder-like smear with high-molecular-weight protein in a western blot using **HA antibody** (Genscript, A01963) after immunoprecipitation with HA antibody (Genscript, A01244). Secondary antibody: Anti-rabbit.

The quantitation data of ubiquitinated proteins were presented in Fig.2g. Our data showed a significant increased ubiquitinated signal in the wild type expressing *CRK1:HA*, than the *fbp1Δ* mutant expressing the *CRK1:HA*. MG132 treated cells had stronger signal than without MG132 treatment.

Fig.2g. Quantitative measurements of Crk1-(Ub)_n are cumulative from three independent experiments. Statistical analysis was performed based on Mann-Whitney test. *, $P \leq 0.05$; **, $P \leq 0.01$.

Second, to test whether Crk1 is capable of directly phosphorylating Gpa1, they purified Crk1:HA and Gpa1:FLAG and performed an *in vitro* kinase assay. They observed the slower migrating form of Gpa1 in the presence of both ATP and Crk1:HA (Fig. 4g), indicating that Crk1 directly phosphorylates Gpa1.

Again there is a heavy reliance on shifted bands, in one case it is inferred to be ubiquitin and the other it is inferred to be phosphorylation.

A: We noticed the shifted bands in our western blot and made hypothesis based on the function of proteins. For Crk1 ubiquitination, we detected a ladder-like smear with high-molecular-weight protein, but not a single shifted band (Fig. 2g; Image 1-Image 3), in a western blot, which suggests a PTM due to poly-ubiquitination. We also used ubiquitin antibody and detected the ubiquitin signal (Image 1). Therefore, we are confident that this smear is ubiquitinated Crk1.

For Gpa1 phosphorylation, we thank reviewer for the constructive suggestion and have performed phosphatase assay (Fig. 4f) and *In vitro* kinase assay (Fig. 4g). Our data confirm that the Gpa1 band shift is a result of its phosphorylation.

*** I do not agree that this experiment is detecting phosphorylation of Gpa1-Flag since a prominent Gpa1-Flag band is observed in the lane lacking Gpa1-Flag. ***

A: We apologize for the wrong information in our previous submission. It was a labeling error. We have corrected it. The first lane is Gpa1:FLAG plus ATP without Crk1, the second lane is Gpa1:FLAG plus Crk1 without ATP. We provided the original label in our X-ray film (Image 4), the record (Image 5), and documentation of reproducibility (Image 6) for your information. We carefully double-checked all figures and information to avoid similar mistakes. We sincerely apologize for this mistake.

Image 4. Original X-ray film for the figure 4g.

Image 5. Original record for the figure 4g.

Image 6. Documentation of reproducibility. We repeated this experiment multiple times with similar outcome. We showed here two other images from independent experimental repeats.

This phosphorylation experiment is lacking other essential controls. I would like to see Gpa1 plus ATP without Crk1, or better yet with catalytically inactive Crk1 mutant, to prove that it is purified and activated Crk1 that is the kinase and not a contaminant kinase.

A: We corrected Fig.4g and the first lane is the control that Gpa1:FLAG plus ATP without Crk1.

We enriched the Gpa1:FLAG protein from *Cryptococcus* strain and did phosphatase assay before setting up *In vitro* kinase assay.

We agree that it would be ideal to have a catalytically inactive Crk1 mutant as a negative control. We have attempted to generate such a strain in this revision, but it appeared to be difficult and takes time to identify the site of kinase activity. We will continue working on such mutants in the future study. Because we did not observe the slower migrating band of Gpa1 in the control that Gpa1:FLAG plus ATP without Crk1, we are confident with the conclusion that Crk1 can phosphorylate Gpa1.

Also since purified protein is being used the authors should show evidence of purity, by Coomassie staining of purified Crk1 and Gpa1.

A: Sorry for the missing information. We purified Gpa1:FLAG via immunoprecipitation with FLAG antibody or Crk1:HA with HA antibody. The purified proteins were resolved by 10% SDS-PAGE and visualized with Coomassie blue staining (Image 7). We showed here two images from independent experimental repeats. Arrows indicate Gpa1:FLAG or Crk1:HA band.

Image 7. Coomassie blue staining to show purified Gpa1:FLAG and Crk1:HA band. Gpa1:FLAG was eluted with 3x FLAG peptide or SDS-PAGE sample buffer. Crk1:HA was eluted with SDS-PAGE sample buffer. Red arrow indicates Gpa1:FLAG band. Orange arrow indicates Crk1:HA band.

Again, I would like to see documentation of reproducibility and quantitation of abundances of the modified and unmodified protein.

A: Please see our documentation of reproducibility (Image 6) for in vitro kinase assay and the quantitation data (Image 8).

Image 8. The quantitation data of *In vitro* kinase assay. *, $P \leq 0.05$.

Reviewer #3 (Remarks to the Author):

I thank the authors for their effort to address the comments raised by this reviewer. Overall, I found the revised manuscript substantially improved. I applaud the authors for their clear presentation of the data and for sharing data in ways that are usable by others (Table S1). Overall, the work contributes significantly to an improved understanding of the molecular mechanisms governing titan cell formation and pathogenesis. Despite this there are still some issues that need to be addressed before the manuscript is suitable for publication.

Panel 3C CRK2deltaPEST is not discussed. I suggest adding it at line 205.

A: Thank you for the suggestion, we added it in this revision.

Supplemental figures are well presented and contribute to understanding of the work.

Supplemental Tables need clear, descriptive legends.

A: We added table legends in this revision.

It is difficult to see how statements in line 267-269 suggesting that exogenous cAMP increases titan cell production in all strains are supported by the data presented in figure S6b, were there is no significant difference in median cell size for any strains except *crk1*. It would be better to present these data as scattered points, similar to S6c.

A: Thank you for the suggestion. We found that exogenous cAMP increases titan cell percentage in all strains (figure 4a), but significantly increased median cell body size only in the *crk1Δ* mutant (figure s6c). We corrected this sentence to “We observed that the addition of exogenous cAMP significantly increased titan cell production with increased DNA content in the *crk1Δ* mutant”. We presented S6b using scattered points, similar to S6c.

Is Figure S6D (top and bottom panels) not discussed in the text? This figure is very confusing. The referenced data is apparently the two panels floating off to the side on the right? This layout needs to be improved for clarity. Also, it's unclear what the blue lines in plot WT+cAMP indicate, as these do not match 1C/2C peaks and lack labels.

A: We thank the reviewer for these suggestion on figure S6d. This figure was used to address reviewer1's comment #7 “all of the strains in Fig 4 that are putatively producing titan cells

should be analyzed for ploidy to confirm titan cell formation in this analysis and the data presented in the figure”. In this revision, we reorganized figure S6d to S6c, S6d, and S6h to make the data clear. We also discussed each figure in the text: “We observed that the addition of exogenous cAMP significantly increased titan cell production with increased DNA content in the *crk1Δ* mutant (Fig. 4a and Supplementary Fig. 6b-6c)”, “Consistent with this observation, the *crk1Δ* mutant produced significantly fewer titan cells than H99 or the *crk1Δ+CRK1* strain when *GPAI* was overexpressed (Fig. 4k and Supplementary Fig. 6h)”

We improved the image of DNA content.

We are sorry for the confusion in data presentation. The three blue lines indicate the 1C/2C/4C peaks, which are same as lines in figure 1e and 3c. Because of limited space, we only add blue lines in plot WT+cAMP, but not other plots. Here, the peak shift between red line and black line indicates the increased DNA content of titan cells, so we deleted the blue lines in this revision.

Data shown in Figure 4i,j S6g significantly improve the manuscript.

A: Thanks.

Figure 4K: it would be better to include the *GPA1OE* genotype on this figure panel.

A: Yes. We included “+*P_{GDP1}-GPAI*” on figure 4k.

In figure 4I, it is disingenuous to change mutant names to "typical" or "titan", especially since these terms are used for subpopulations throughout the rest of the manuscript. Were these cells fractionated by size? If not, then simply report strain names and explain the logic to your reader. Please make this correction in the text at lines 332, 334 as well.

A: Thank you for the suggestion. We didn't fractionate cells by size, so we corrected the typical cells to “the *crk1Δ P_{GDP1}-Gpa1:FLAG* strain” and titan cells to “*CRK1^{ΔPEST} P_{GDP1}-Gpa1:FLAG* cells” in the text and correct the typical to “*crk1Δ GPAI^{OE}*” and titan cells to “*CRK1^{ΔPEST} GPAI^{OE}*” in figure 4I.

Symbols on Figure 5a don't reproduce well in black and white. It is hard to differentiate WT and CRK1dPEST.

A: Thank you for the suggestion. We changed symbols on figure 5a.

Figure 5B is not quantitative, but is described in the text to show that particular strains showed an "increased presence". This must be corrected by presenting quantitative data, as has been shown in other points in the manuscript.

A: We added the quantitative data of titan cell percentage in figure 5b.

Line 365/366, 452-454, 455-458, Multiple other authors have already shown that increased titan cell production correlates with diminished fungal virulence, as has already been pointed out to the authors. Statements that over emphasize the novelty of the authors findings undermine the overall work and should be revised. Others who have done work supporting this conclusion, in addition to work on Pdr802, should be cited, as suggested previously.

A: We cited multiple works to show that increased titan cell production correlates with **increased** fungal virulence. The work on Pdr802 is the only one reporting that titan cell production diminishes fungal virulence before our work on Fbp1 and Crk1.

There are still some typos that need to be corrected, and some which have been incorrectly introduced in the new version, perhaps on the advice of others. I can confirm that I have a degree in English writing and grammar (in addition to being a "native" speaker) and that these suggestions are grammatically correct:

A: We are grateful for the careful edits and the recommended modifications by the reviewer. We corrected these typos as recommended.

Line 56 "elongated hyphae cells" change to: elongated hyphal cells

Line 63 "cellular heterogeneity ... are often observed" change to: cellular heterogeneity ... is often observed

Line 81 "support fine-tune cell..." change to: support fine-tuned cell size regulation

Line 92 "mutant persists low levels.." change to: mutant persists at low levels...

Line 95 "low pulmonary persistency...." change to: low pulmonary persistence

Line 101 "The Gpa1-cAMP pathway activation": "The" is not needed, though this is mostly stylistic.

Line 105/106: "the Gpa1 G protein signal pathway through a kinase regulator Crk1" change to: the Gpa1 G protein signalling pathway through a kinase regulator, Crk1.

Line 119: lower case w in "we"

Line 148: I think "support" is a more accurate word to use than "perform"

Line 288-289 "is accounted" change to: To determine whether Crk1 activity accounts for Gpa1 phosphorylation....

Line 296: "proteins from the H99..." change to: proteins from H99...

A: We have corrected all the typos.

Line 305: this needs to be two separate sentences and the second sentence needs correction:
We isolated the total protein from H99 expressing Gpa1:FLAG that were cultured in YPD overnight. The sample was treated with protein phosphatase to remove phosphorylation from Gpa1 before purification.

I suggest rewriting more simply:

Total protein was collected from overnight YPD cultures of H99 expressing Gpa1:FLAG and treated with protein phosphatase to remove phosphorylation before purification. We observed the slower migrating form of Gpa1 only in fractions treated subsequently with both ATP and Crk1:HA (Fig. 4g), indicating that Crk1 directly phosphorylates Gpa1.

A: We have modified as suggested.

Line 309-311: "Total proteins... were isolated" change to: Total protein ... was isolated (yes, English is weird)

Line 315: "or a rich medium of YPD" change to: "rich medium (YPD)" or "YPD rich medium"

Line 337: comma, not semi colon.

Line 339: "levels... and ...production is still" change to: levels... and ...production are still

Line 342: "in vivo and mice" change to: "in vivo, and mice"

Line 350: "infection with Crk1OE strain" change to: infection with the Crk1OE strain

Line 351: "in infected lungs sections" change to: in infected lung sections

Line 352: "the lung of mice..." change to: the lungs of mice

Line 429: "functions upstream of the Gpa1-" change to: functions upstream of Gpa1-

Line 469: "has been reported to both stimulates and inhibits" change to "stimulate and inhibit"

Line 1109: "proportion in cells overexpression of GPA1" change to: proportion of cells overexpressing GPA1"

A: We have corrected all the typos.

Figure legends also need to be read carefully for grammatical issues.

A: We have edited figure legends for grammatical correction.

Reviewer #2 (Remarks to the Author):

I am satisfied with the revisions, thank you for the response.

Reviewer #3 (Remarks to the Author):

Overall the authors have addressed my concerns.

However, I refer the authors to the work of Gish et al., (2016) mBio, where deletion of USV101 resulted in strains with reduced dissemination to the brain not explained by the capsule-related phenotypes. Dambuza et al., and Hommel et al., (2018) PLOS Pathogens later demonstrated that USV101 is a major negative regulator of titan cell formation, and were the first to conceptually link increased titan cell formation to reduced virulence.

NCOMMS-21-47756

Response to reviewers' comments

REVIEWERS' COMMENTS

Reviewer #2 (Remarks to the Author):

I am satisfied with the revisions, thank you for the response.

A: Thanks.

Reviewer #3 (Remarks to the Author):

Overall the authors have addressed my concerns.

However, I refer the authors to the work of Gish et al., (2016) mBio, where deletion of USV101 resulted in strains with reduced dissemination to the brain not explained by the capsule-related phenotypes. Dambuza et al., and Hommel et al., (2018) PLOS Pathogens later demonstrated that USV101 is a major negative regulator of titan cell formation, and were the first to conceptually link increased titan cell formation to reduced virulence.

A: We have included these references in our discussion on page 15.